# UltraLED: Learning to See Everything in Ultra-High Dynamic Range Scenes

**Yuang Meng**[*]  **Xin Jin**[*]  **Lina Lei**  **Chun-Le Guo**[†]  **Chongyi Li**
VCIP, CS, Nankai University
https://srameo.github.io/projects/ultraled

## Abstract

Ultra-high dynamic range (UHDR) scenes exhibit significant exposure disparities between bright and dark regions. Such conditions are commonly encountered in nighttime scenes with light sources. Even with standard exposure settings, a bimodal intensity distribution with boundary peaks often emerges, making it difficult to preserve both highlight and shadow details simultaneously. RGB-based bracketing methods can capture details at both ends using short-long exposure pairs, but are susceptible to misalignment and ghosting artifacts. We found that a short-exposure image already retains sufficient highlight detail. The main challenge of UHDR reconstruction lies in denoising and recovering information in dark regions. In comparison to the RGB images, RAW images, thanks to their higher bit depth and more predictable noise characteristics, offer greater potential for addressing this challenge. This raises a key question: *can we learn to see everything in UHDR scenes using only a single short-exposure RAW image?* In this study, we rely solely on a single short-exposure frame, which inherently avoids ghosting and motion blur, making it particularly robust in dynamic scenes. To achieve that, we introduce UltraLED, a two-stage framework that performs exposure correction via a ratio map to balance dynamic range, followed by a brightness-aware RAW denoiser to enhance detail recovery in dark regions. To support this setting, we design a 9-stop bracketing pipeline to synthesize realistic UHDR images and contribute a corresponding dataset based on diverse scenes, using only the shortest exposure as input for reconstruction. Extensive experiments show that UltraLED significantly outperforms existing single-frame approaches. Our code and dataset are made publicly available at https://srameo.github.io/projects/ultraled.

## 1 Introduction

UHDR scenes are common in daily life, such as night streets illuminated by headlights and streetlights, or indoor environments with bright window light and dim interiors. The extreme contrast in these scenes exceeds the capacity of conventional cameras in a single exposure, often resulting in bimodal intensity distributions, where highlight and shadow regions form separate peaks with little midtone. This phenomenon makes it difficult for cameras to retain details across the full brightness range, often causing loss of visual information and reduced image fidelity.

Recent methods often use bracketing [17, 14, 56, 7, 36] to capture a sequence of exposures ranging from short to long. Nevertheless, this kind of approach inherently suffers from inter-frame motion, resulting in misalignment and ghosting artifacts that degrade reconstruction quality. Some generative approaches [30] hallucinate missing highlights from single long-exposure input (i.e., Fig. 1(a)), yet

---

[*]Equal Contribution.
[†]C. L. Guo is the corresponding author.

39th Conference on Neural Information Processing Systems (NeurIPS 2025).

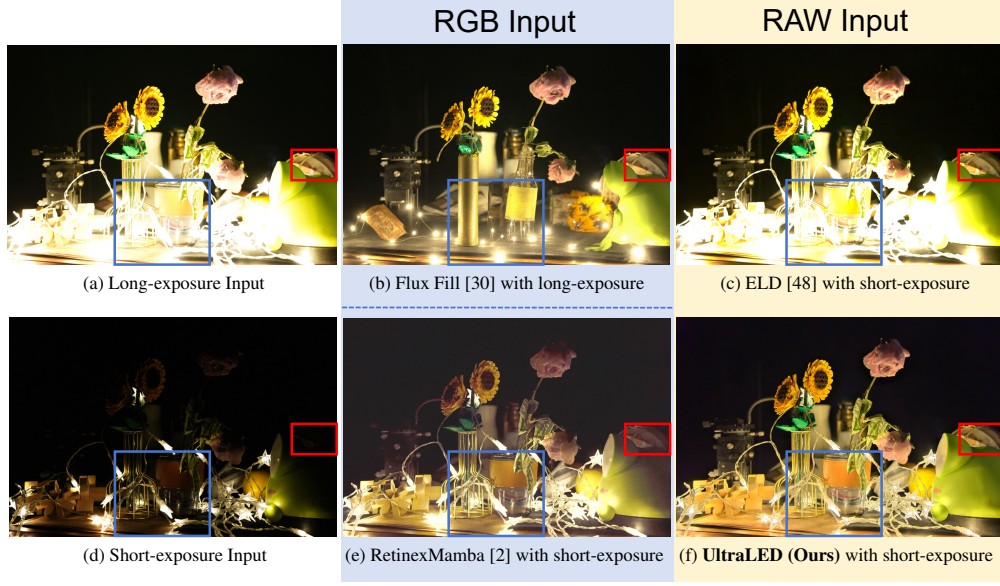

Figure 1: Visualization results of different methods for UHDR scene reconstruction.

struggle to recover realistic details in saturated regions as shown in Fig. 1(b). In addition, some methods [15, 2] adopt short-exposure inputs (i.e., Fig. 1(d)) to avoid highlight clipping. Nevertheless, due to the limited dynamic range of 8-bit RGB images, shadow regions may be represented with as few as 4 bits, causing severe quantization artifacts and noise, as shown in Fig. 1(e).

The observations in Fig. 1 lead to two critical conclusions: *1) long-exposure-based methods fundamentally fail to recover highlight regions in the absence of short-exposure references;* and *2) short-exposure-based methods are primarily limited by bit depth constraints and noise in dark regions.* Fortunately, RAW images can offer substantial advantages for such tasks due to their higher bit depth and simpler noise distributions. As shown in the red box of Fig. 1(c), the globe on the right, which is in darkness, has been restored to a result similar to that of a long exposure. The results show that RAW images can preserve more complete and realistic details even under extremely low-light conditions. Therefore, for short-exposure RAW images, we only need to perform exposure correction, as they already contain both the unclipped highlights and the recovered details from low-light regions. Nevertheless, two core challenges still remain:

- Extreme exposure differences compounded with severe noise in low-light regions result in a highly ill-posed reconstruction problem. Jointly optimizing exposure correction and denoising degrades both learning efficiency and reconstruction accuracy.

- In the RAW domain, complex noise models [48] demonstrate superior effectiveness under extremely low-light conditions, whereas simpler models such as the PG noise model perform better in brighter regions [28]. This discrepancy stems from the fact that the noise distribution tends to follow different known distributions as the image brightness changes. Therefore, it is challenging to simultaneously reconstruct the bright and dark areas of UHDR scenes.

To address these challenges, we decouple the UHDR reconstruction process into two distinct stages. First, leveraging the high bit-depth property of RAW images, we train a network to generate a ratio map that dynamically corrects local exposure across the image. Second, recognizing that noise characteristics vary with brightness, we introduce a brightness-aware noise model. To guide the denoising network, we encode brightness information using a ratio map, enabling joint reconstruction of both bright and dark regions.

In addition, there is no publicly available dataset suitable for training and evaluating UHDR reconstruction methods. To bridge this gap, we develop a novel pipeline that leverages the characteristics of RAW images to synthesize a 9-stop exposure stack, from which we separately obtain noise-free ratio maps and the corresponding fused results. Based on this pipeline, we construct a high-quality

dataset tailored for UHDR reconstruction. Notably, all scenes are captured under static conditions, enabling multi-exposure acquisition without alignment issues or ghosting artifacts.

Our main contributions are summarized as follows:

- We propose a novel method for reconstructing UHDR scenes using only a single-frame RAW image, in which a two-stage pipeline is proposed to decouple exposure correction from denoising, fully leveraging the properties and advantages of RAW data.

- We propose a brightness-aware noise model and a ratio map encoding scheme that synergistically guide the network in recovering fine details across varying exposure levels.

- We design a new data pipeline for multi-exposure fusion and contribute a corresponding dataset for benchmarking UHDR reconstruction performance.

## 2   Related Work

**RAW Denoising.**   Since the pioneering work of SIDD [1], the potential of RAW data for low-light imaging has been extensively explored. Recent studies simulate low-light conditions by degrading normal-light images. SID [10] introduces a real-world RAW dataset to train denoising networks. Subsequent approaches have sought to model noise more precisely via either physically calibrated parameter estimation [48, 54] or network-based learning [9, 40, 28], achieving improved denoising performance. However, existing RAW denoising methods [10, 28, 48, 26] cannot adaptively regulate local exposure, rendering them inadequate for UHDR scenarios. In contrast, UltraLED enables pixel-wise brightness adjustment, simultaneously achieving effective denoising and exposure correction.

**Single-image HDR and Low-light Image Enhancement.**   Both single-image HDR reconstruction and low-light image enhancement seek to produce well-exposed images with rich detail [33]. Traditional single-image HDR techniques [5, 3, 4] rely on internal cues to predict scene luminance and extend dynamic range by estimating light source intensity. However, both approaches are fundamentally constrained by the absence of scene information, which is inherently difficult to recover [20, 37, 35]. Single-image HDR methods typically operate on normally exposed inputs [20, 37, 35], requiring generative capabilities to reconstruct the lost details caused by overexposure [56, 52, 24]. This task becomes particularly challenging in cases of severe highlight clipping. Traditional retinex-based low-light enhancement methods [45, 31, 46, 22, 8] decompose an image into reflectance and illumination components, treating the task as illumination estimation. Most low-light enhancement methods [23, 25, 32, 34, 27] operate in the RGB domain, where the limited bit depth, sometimes reduced to only 4 bits in dark regions, significantly limits the fidelity of reconstruction. We address this issue by operating in the RAW domain, where images retain higher bit depth (typically 14 bits). Although certain regions are quantized to 4 bits in RGB, they may preserve up to 10 bits in RAW, allowing for more accurate and faithful recovery. Meanwhile, the high-bit information in the RAW domain preserves a more predictable and simpler distribution of the original noise [10, 48], making it easier to denoise. By denoising directly in the RAW domain, UltraLED surpasses RGB-based approaches [10, 48], achieving superior detail recovery under short exposures.

**Multi-exposure Fusion HDR.**   Traditional approaches typically reconstruct HDR images by merging a sequence of bracketed low dynamic range (LDR) images [38, 17], where the alignment of multiple exposures remains the most challenging step. Several CNN-based methods [29, 50, 49] have achieved improved alignment performance. Peng et al. [41] and FlowNet [19] explore optical flow estimation techniques for more accurate alignment. Given the limited receptive field of CNNs, transformer-based approaches [51, 36] have demonstrated superior performance. More recently, diffusion-based methods [56, 14] have also shown promising results; however, they require substantial computational resources and often produce unrealistic details due to the generative nature of the models. Multi-exposure fusion HDR captures true scene content across exposure extremes but often suffers from alignment errors and ghosting, especially in dynamic or low-light scenes. Single-image methods using long exposure struggle to recover highlight details, while those based on short-exposure RGB inputs face severe quantization and noise in dark regions due to limited bit depth. We address these issues by reconstructing UHDR scenes from a single short-exposure RAW image, achieving better fidelity with lower cost and complexity.

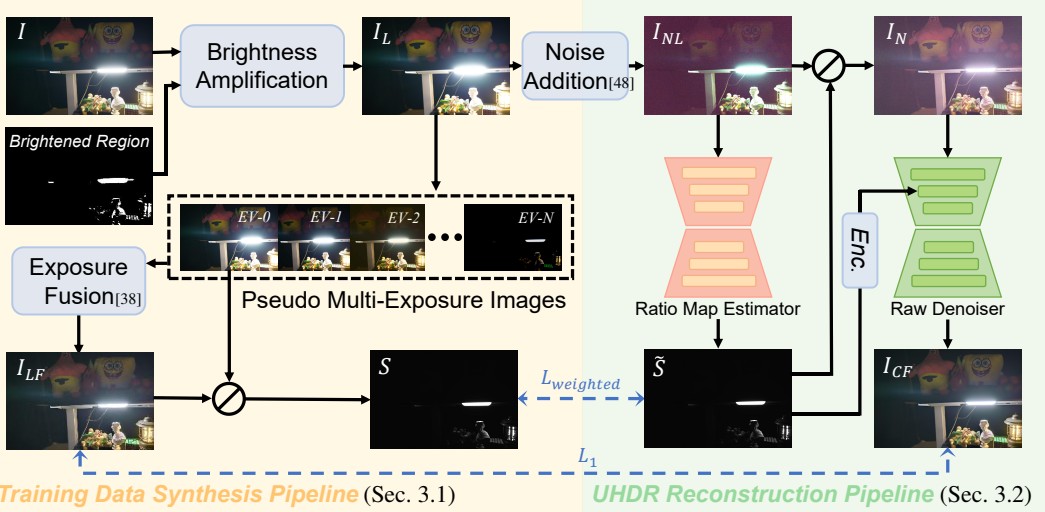

Figure 2: Overview of our framework, which is divided into two parts: **1) Training Data Synthesis Pipeline**: clean and normally exposed image $I$ is used as input. The lighted image $I_L$ is generated by artificially amplifying the brightness in specific regions of $I$. The noise model is then applied to $I_L$ to synthesize the corresponding noisy and overexposed image $I_{NL}$. Meanwhile, $I_L$ is linearly scaled down by different factors and clipped to produce pseudo multi-exposure images from $EV$-$0$ to $EV$-$N$. These images are fused using exposure fusion [38] to obtain $I_{LF}$, and a clean exposure-corrected map (ratio map $S$) is derived by dividing $I_L$ by $I_{LF}$; and **2) UHDR Reconstruction Pipeline**: this pipeline consists of two stages, implemented using two UNet [10]. First, the Ratio Map Estimator takes the noisy and overexposed RAW image $I_{NL}$ as input and outputs the ratio map $\widetilde{S}$. Then, $\widetilde{S}$ is used to correct the exposure of $I_{NL}$, producing a noisy but well-exposed image $I_N$, which is then passed into the RAW Denoiser for denoising guidance. Finally, the RAW Denoiser outputs a clean and well-exposed RAW image $I_{CF}$. Note that, in terms of inference, only the UHDR Reconstruction Pipeline is used.

## 3 Methodology

In this section, we introduce our training data synthesis pipeline and UHDR reconstruction pipeline. An overview of our framework is shown in Fig. 2.

### 3.1 Training Data Synthesis Pipeline

Capturing perfectly aligned, noise-free multi-exposure images under unconstrained, real-world conditions remains highly challenging. Thus, it is non-trivial and often impractical to construct large-scale paired datasets directly from in-the-wild scenes. In this study, we leverage a large-scale repository of well-exposed RAW images to overcome this limitation and synthesize paired data for the whole processing pipeline. RAW images without highlight clipping exhibit a linear relationship between light intensity and signal values, along with predictable noise distributions. This allows us to directly obtain underexposed inputs and multi-exposure sequences, enabling the generation of well-exposed references. Since these well-exposed RAW images are free from highlight clipping, we also simulate artificial overexposure for completeness. Accordingly, our training data synthesis pipeline comprises: 1) brightness amplification, 2) exposure fusion, and 3) noise modeling.

**Brightness Amplification.** To amplify the local image brightness, we artificially synthesize the overexposure, following the procedure of previous work [16, 42, 43] to generate saturated highlight regions. To further simulate the stochastic distribution of highlights in UHDR scenes, we inject additional random highlight patches. Brightened regions are randomly selected, intensified by a stochastic gain factor, smoothed with a bilateral filter, and finally feathered at the boundaries with a Gaussian kernel to simulate light diffusion. Given images that preserve highlight details, we construct pseudo multi-exposure images by progressively reducing global illumination, implemented as linear scaling for RAW data, followed by value clipping. Specifically, for the normalized image, we use the

following formula to synthesize $EV\text{-}i$:

$$EV\text{-}i = \text{clip}(\frac{I_L}{2^i}, 0, 1), \tag{1}$$

where $i$ denotes the downscaling factor of the image's exposure value, $I_L$ denotes the artificially over-exposed RAW input, clip$(x, 0, 1)$ represents clipping the value of $x$ to the range between 0 and 1, simulating overexposure truncation. The resulting exposures, spanning $EV\text{-}0$ to $EV\text{-}N$ (where $EV$ denotes exposure value, with each step doubling or halving incident light), are subsequently used in the HDR fusion stage.

**Exposure Fusion.** Because the pseudo multi-exposure images are synthesized by modulating the brightness of a single highlight-preserved RAW frame, inter-frame misalignment is inherently eliminated. We employ Exposure Fusion [38] to combine the images into a well-exposed composite. Since Exposure Fusion operates in the RGB domain, each RAW frame is first rendered to the RGB domain using predefined ISP parameters, after which the fused result is converted back to the RAW domain via the unprocessing pipeline [6]. This procedure yields a noise-free, well-exposed reference image $I_{LF}$ that has been lighted and fused, as well as a clean ratio map $S$, defined as

$$S = \frac{I_L}{I_{LF}}. \tag{2}$$

**Noise Model.** We divide the noise in UHDR scenes into three parts, including signal-dependent noise, signal-independent noise [48, 54, 21], and **brightness-aware** noise. Our modeling of signal-dependent noise and signal-independent noise is the same as the existing method [48] in the RAW domain. Brightness-aware noise is introduced due to the significant brightness variations across different regions of UHDR scenes and the subsequent exposure correction process. Our final noisy input $I_N$, representing the UHDR RAW image after exposure correction, is defined as:

$$\begin{aligned} I_L &= \text{SA}(I), \\ I_{NL} &= I_L + N_{de} + N_{in}, \\ I_N &= I_{NL} + N_{ba}, \end{aligned} \tag{3}$$

where $I$, $I_{NL}$, $N_{de}$, and $N_{in}$ denote the well-exposed and noise-free RAW image, the UHDR image, the signal-dependent noise, and the signal-independent noise, respectively. $N_{ba}$ refers to the brightness-aware noise. SA denotes the processes of synthesizing artificial overexposure regions. The detailed formulation for synthesizing the final noisy image is given by:

$$I_N = \hat{\text{SA}}^{-1}\left(\left(\mathcal{P}\left(\frac{\text{SA}(I)}{R \cdot K}\right) \cdot K + N_{in}\right) \cdot R\right), \tag{4}$$

where $\hat{\text{SA}}^{-1}$ denotes the processes of correcting the exposure using the ratio map estimator network, thereby modeling the $N_{ba}$ component of noise. $K$ is the random system gain for signal-dependent noise (shot noise), and $R$ is a random attenuation ratio for brightness.

For most RAW image denoising methods, in order to ensure numerical stability, the noisy image should be scaled to the original range. Up to this point, we have obtained all the reference images used for training, including the UHDR RAW image after exposure correction $I_N$ as input, the ratio map $\widetilde{S}$ for exposure correction, and the noise-free and well-exposed ground truth image $I_{LF}$.

### 3.2 UHDR Reconstruction Pipeline

**Decoupling Exposure Correction and Denoising.** We decouple the reconstruction of UHDR scenes into two parts: learning the exposure correction pattern of the image and the simple noise distribution, respectively. Specifically, the pipeline consists of three steps: 1) Using a simple UNet that has the same architecture as that used in SID [10] to predict the ratio map $\widetilde{S}$; 2) The predicted ratio map $\widetilde{S}$ is then applied to our UHDR input $I_{NL}$ to construct a well-exposed noisy input $I_N$; 3) Finally, another same UNet is utilized to denoise $I_N$.

**Ratio Map Encoding.** The ratio map serves as a noise-free representation of image brightness. Our objective is to introduce an implicit feature capturing the relationship between brightness and noise.

Regions with similar brightness levels tend to have similar ideal ratio values and correspondingly similar noise intensity levels. Since low-dimensional ratio values are insufficient to represent the complex noise distribution across varying brightness conditions, we expand the dimensionality of the ratio map using Gaussian encoding. This is because Gaussian distributions can model the gradual change in noise intensity as ratio values become closer. Additionally, we introduce a weight corresponding to the ratio before encoding, which modulates the effective signal strength. For each ratio $r$, the encoding $EC_r$ is as follows:

$$EC_r = \frac{\exp\left(-\frac{(r-y)^2}{2\sigma^2}\right)}{\sqrt{2\pi} \cdot \sigma \cdot r}, \tag{5}$$

where $\sigma$ is an externally defined hyperparameter, and $y$ is a vector representation obtained by uniformly sampling the ratio parameter space at a predefined dimensionality. The ratio parameter space should encompass all possible values that the ratio can take. The corresponding $r$ of each pixel of the ratio map $\widetilde{S}_i$ is calculated as follows:

$$r = \frac{R}{\widetilde{S}_i}, \tag{6}$$

The integrated encoding scheme is injected into the denoising network through a multi-layer perceptron and zero-initialized convolutional residual block, which enables ratio-adaptive denoising guidance to better restore the details and textures under different ratios. This formulation ensures the denoiser dynamically adjusts its operation based on amplification factors while maintaining noise suppression consistency across varying ratio conditions. More derivation and experiments can be found in the supplementary material.

**Loss Functions.** For our two-stage network, we employ different loss functions for different stages. In the exposure correction stage, due to the large range of values of our ratio map, we use a weighted L1 loss [39] to ensure numerical stability instead of a standard L1 loss:

$$\mathcal{L}_{weighted} = \frac{1}{N} \sum_{i=1}^{N} \frac{|S_i - \widetilde{S}_i|}{S_i + \epsilon}, \tag{7}$$

where $\epsilon$ is the numerical stability coefficient. The loss function for the denoising stage is a standard L1 loss: $\mathcal{L}_1 = \frac{1}{N} \sum_{i=1}^{N} |I_{LF} - I_{CF}|$.

## 4 Experiments

### 4.1 Proposed UHDR Dataset

To systematically assess the reconstruction performance of various methods on UHDR scenes, we construct a UHDR image dataset comprising RAW images and their corresponding RGB counterparts. The thumbnail of our UHDR dataset is shown as Fig. 3.

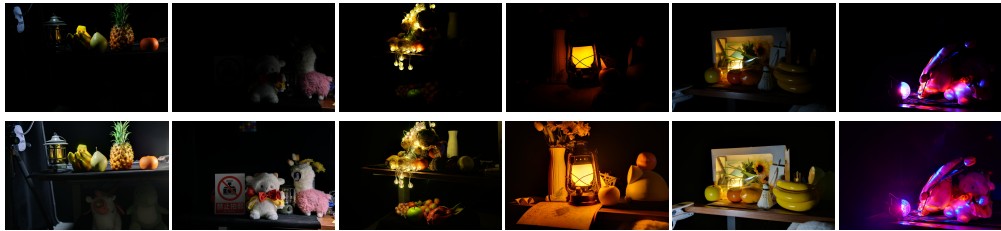

Figure 3: A thumbnail of our UHDR dataset (convert to RGB images for visualization), where the images in the top row are UHDR inputs and the images in the bottom row are ground truth. Our UHDR dataset comprises 24 UHDR scenes, the majority of which are captured under three lighting conditions (ratio ×50, ×100, and ×200). For each combination of lighting condition and ISO setting, we captured three RAW images of the same scene to ensure statistical reliability. In total, the dataset includes 585 paired RAW images for evaluation.

To eliminate interference from ambient illumination, paired data are captured in a controlled indoor environment using a remotely operated Sony A7M4 camera. The experimental setup includes multiple types of light sources and diverse object arrangements, allowing directly illuminated objects to coexist with those in shadow, thereby simulating real-world UHDR scenes. All surfaces within the shooting area, including tables and walls, are covered with anti-reflective fabric to minimize secondary reflections, ensuring that scene luminance is governed almost exclusively by direct illumination. For each scene, several sets of noisy images are recorded at different ISO settings according to ratio gradients of 50, 100, and 200. Unlike conventional low-light denoising datasets [10, 48], here the "ratio" denotes the amplification factor required to achieve ideal exposure in the darkest region. High-quality reference images are obtained by capturing nine exposure values ($EVs$) at the lowest ISO with the corresponding longest exposure time, and synthesizing noise-free and well-exposed ground truth via Exposure Fusion [38].

Notably, in contrast to typical HDR scenes, the illuminance of dark regions in UHDR dataset can be as low as 0.001–0.2 lux, whereas bright regions reach 0.3–100 lux. Consequently, in some scenes, maintaining proper exposure for the highlights forces the RGB pixel values (8 bit) in shadow areas to collapse to the range 0–4. Under such extreme conditions, RGB images are incapable of recovering shadow details, and UltraLED therefore outperforms RGB-based methods by a substantial margin.

## 4.2 Implementation Details

**Training Data and Network Architecture.** We train our network using the synthesized data based on the ground truth of the RawHDR dataset [57], selected for its high bit depth, good exposure, and noise-free characteristics. Our ratio map estimator employs a simple U-Net architecture [44] to capture the characteristics of the brightness distribution. This module takes a noisy RAW image as input and outputs a noise-free ratio map. The hyperparameter $\sigma$ used to construct the ratio map encoding is set to 30 in our experiments. For the denoising network, various architectures can be adopted. However, since the strength of our approach lies primarily in the overall pipeline and data processing strategy, we use the same simple U-Net framework [44] for a fair comparison with other methods. Note that employing more complex networks with higher computational capacity and parameters could further enhance performance. The related experiments are discussed in the supplementary material.

**Optimization Strategy and Evaluation Metrics.** We first train the ratio map estimator network and then freeze its parameters before training the denoising network. For the denoising network, we adopt a widely used training strategy for RAW domain denoising, as described in ELD [48, 28]. The ratio map estimator requires relatively fewer training iterations, only 30,000 iterations using the Adam optimizer, with a learning rate of $5 \times 10^{-5}$. To comprehensively evaluate quantitative performance, we employ PSNR, SSIM, and LPIPS as evaluation metrics.

## 4.3 Comparison Experiments

We conduct both quantitative and qualitative evaluations on the UHDR dataset. Additionally, we observed that some low-light datasets, such as the SID dataset [10], often unintentionally capture UHDR scenes during outdoor night photography. Previous methods [48] have struggled to accurately restore such scenes. To assess the generalization capability of our approach, we also perform qualitative evaluations on such scenes from the SID dataset.

To facilitate a comparison that better reflects actual visual perception, we convert all outputs to the RGB domain following the same ISP process. To ensure fair evaluations, we align the exposure levels of different methods' results using the approach proposed by RAWNeRF [39], thereby eliminating brightness variation as a confounding factor in image quality assessment. We then conduct comprehensive comparisons between UltraLED and different approaches including low-light enhancement, single-image HDR, as well as RAW domain denoising and fusion methods. It is worth noting that we standardize the exposure of all RAW inputs using the commonly adopted method for RAW domain denoising [10, 48], which calculates exposure ratios based on the ISO and shutter speed of the images. The inputs of different methods are as follow. **1) Short-exposure RAW input**: UltraLED, RAW domain-based denoising followed by fusion methods including PG, ELD [48], and LED [28]. **2) Short-exposure RGB input**: Low-light image enhancement methods including RetinexMamba [2],

EnlightenGAN [25], ZeroDCE [23], and Kind [55]. **3) Long-exposure RGB input**: Single-image HDR methods including HDRUNet [12], HDRTVNet [13], and HDRTVNet++ [11].

**Quantitative Results.** As shown in Tab. 1, Tab. 2, and Tab. 3, UltraLED demonstrates superior performance in handling UHDR scenes. In Tab. 1, single-image HDR methods [12, 13, 11] use long-exposure RGB images as input. The extended exposure time leads to severe overexposure and clipping, and due to the lack of prior information, these HDR methods are unable to recover realistic details in heavily overexposed regions. This results in significant structural discrepancies between the outputs and the ground truth, reflected in low SSIM scores. However, since the inputs are noise-free, the PSNR remains relatively high. In Tab. 2, RAW domain denoising methods [48, 28] first process inputs with varying amplification ratios and then fuse the outputs using the same fusion method [38] adopted in our data synthesis pipeline. While this strategy yields high SSIM because of the structural similarity with the ground truth, it fails to account for the noise variation across different exposures, resulting in noise mixing and performance degradation. Additionally, the need for multiple denoising and fusion steps significantly increases their computational costs compared to ours. In Tab. 3, compared to low-light image enhancement methods [25, 23, 55, 2] operating in the RGB domain, UltraLED leverages the higher bit depth of RAW images, providing more comprehensive information. Furthermore, our decoupling strategy and brightness-aware noise modeling enable more effective denoising and detail restoration. We also conducted a user study to compare UltraLED with other methods [48, 23, 2] across different scenes and sensors. The results of this study are provided in the supplementary materials.

Table 1: Comparison with HDR methods. HDR methods [12, 13, 11] take a long-exposure image as input. Although such input is noiseless, it suffers from highlight clipping in bright regions. The best result is marked in red.

| Methods | HDRTVNet | HDRTVNet++ | HDRUNet | *UltraLED (Ours)* |
|---|---|---|---|---|
| PSNR | 25.318 | 25.344 | 25.341 | 27.591 |
| SSIM | 0.830 | 0.828 | 0.801 | 0.898 |
| LPIPS | 0.095 | 0.095 | 0.104 | 0.075 |

Table 2: Comparison with RAW domain methods. LED [28], PG, and ELD [48] amplify the input image by different ratios for denoising, and then fuse the results using the same method [38] as producing ground truth. The best result is marked in red.

| Methods | LED | PG | ELD | *UltraLED (Ours)* |
|---|---|---|---|---|
| PSNR | 25.879 | 25.142 | 25.942 | 27.591 |
| SSIM | 0.888 | 0.878 | 0.886 | 0.898 |
| LPIPS | 0.083 | 0.101 | 0.081 | 0.075 |

Table 3: Comparison with low-light image enhancement methods. The best result is marked in red.

| Methods | ×50 | | | ×100 | | | ×200 | | |
|---|---|---|---|---|---|---|---|---|---|
| | PSNR | SSIM | LPIPS | PSNR | SSIM | LPIPS | PSNR | SSIM | LPIPS |
| EnlightenGAN | 21.154 | 0.558 | 0.548 | 19.833 | 0.512 | 0.605 | 18.878 | 0.464 | 0.653 |
| Zero-DCE | 21.920 | 0.534 | 0.617 | 20.443 | 0.463 | 0.673 | 19.277 | 0.400 | 0.757 |
| Kind | 20.191 | 0.458 | 0.748 | 18.907 | 0.356 | 0.832 | 17.538 | 0.282 | 0.888 |
| RetinexMamba | 22.770 | 0.705 | 0.193 | 21.587 | 0.641 | 0.229 | 20.600 | 0.586 | 0.271 |
| *UltraLED (Ours)* | 27.591 | 0.898 | 0.075 | 27.327 | 0.887 | 0.095 | 27.091 | 0.860 | 0.121 |

**Qualitative Comparison.** As shown in Fig. 4, UltraLED outperforms existing approaches in UHDR scenes by achieving superior denoising and detail restoration in low-light areas, along with better color fidelity in bright regions. Fig. 5 further demonstrates that UltraLED can simultaneously correct exposure and denoise, effectively recovering textures even in severely overexposed regions (after amplification via ratio) where conventional RAW domain denoising methods [48] fail. These results validate the generalizability of our approach across different camera models and outdoor scenes. More visualization results can be found in the supplementary material.

## 4.4 Ablation Studies

In this section, we provide ablation studies to demonstrate the effectiveness of our pipeline and different modules. The visualization results of ablated models are presented in the supplementary material.

**RAW Domain Decoupling Strategy.** To evaluate the effectiveness of our decoupling strategy for exposure correction and denoising, we compare UltraLED with the one-stage method that

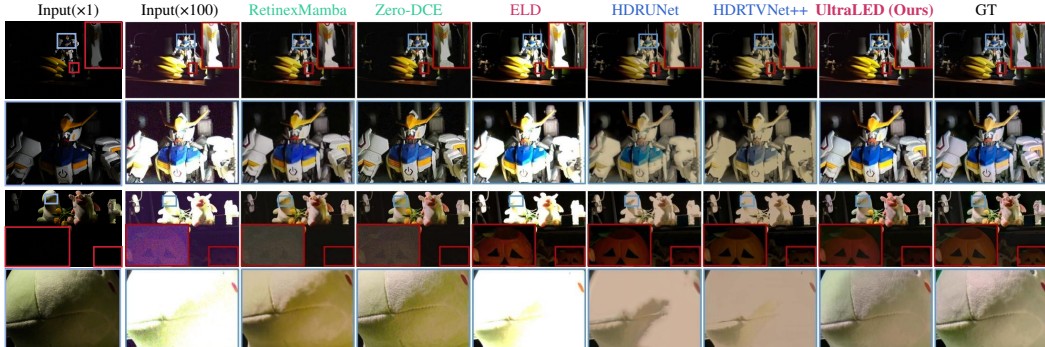

Figure 4: Visualization results of different methods on the UHDR dataset. The methods labeled blue take a long-exposure noiseless image as input, the methods labeled red take a short-exposure RAW image as input, and the methods labeled green take a short-exposure image RGB as input. UltraLED achieves good visual effects in both bright and dark regions. Note that there may be slight differences in our tone because our training data underwent a reversed ISP process. However, this difference is generally negligible and aligns with real-world conditions.

also operates in the RAW domain under identical settings. Additionally, to highlight the inherent advantages of the RAW domain in restoring UHDR scenes, we conduct experiments using the same settings in the RGB domain. The results are presented in Tab. 4.

**Brightness-Aware Noise and Ratio Map Encoding.** To assess the contributions of the proposed brightness-aware noise modeling and ratio map encoding, we conduct ablation studies on these two modules. The results, as shown in Tab. 5, demonstrate that both components significantly enhance denoising performance and detail recovery in UHDR scenes.

Table 4: Ablation on the image domain and the application of the decoupling strategy.

| | Ratio | ×50 | ×100 | ×200 |
|---|---|---|---|---|
| Domian | Decoupling | PSNR/SSIM | PSNR/SSIM | PSNR/SSIM |
| RGB | ✗ | 22.844/0.693 | 21.732/0.622 | 20.531/0.569 |
| RGB | ✓ | 22.937/0.688 | 21.560/0.613 | 20.376/0.554 |
| RAW | ✗ | 25.137/0.878 | 24.896/0.866 | 24.632/0.847 |
| RAW | ✓ | **27.598/0.898** | **27.327/0.887** | **27.091/0.860** |

Table 5: Ablation on $N_{ba}$ and Ratio Map Encoding.

| | Ratio | ×50 | ×100 | ×200 |
|---|---|---|---|---|
| $N_{ba}$ | Encoding | PSNR/SSIM | PSNR/SSIM | PSNR/SSIM |
| ✗ | ✗ | 26.012/0.862 | 25.814/0.838 | 24.513/0.761 |
| ✗ | ✓ | 26.301/0.878 | 26.005/0.849 | 24.678/0.766 |
| ✓ | ✗ | 27.062/0.882 | 26.893/0.871 | 26.734/0.851 |
| ✓ | ✓ | **27.598/0.898** | **27.327/0.887** | **27.091/0.860** |

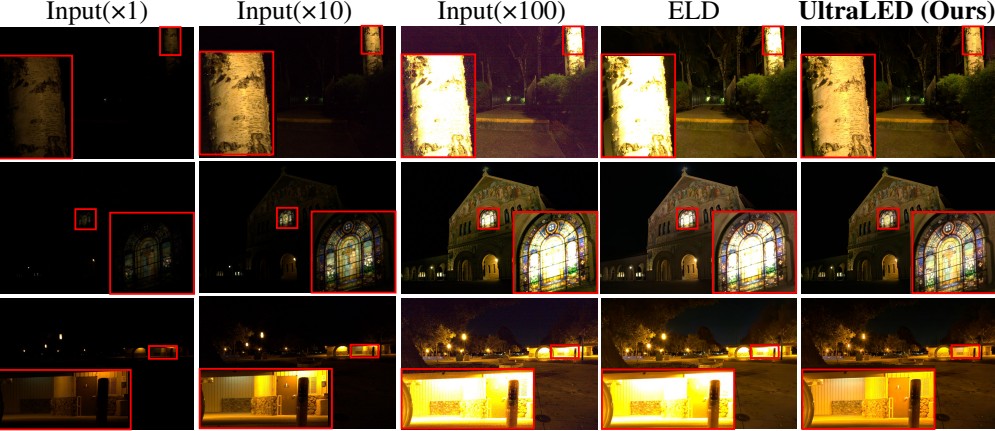

Figure 5: Visual comparison on the SID [10] dataset. It is well-known that RGB-based methods [55, 23, 25, 2] have limited ability to recover details in extremely low-light regions. Therefore, we only compare UltraLED with ELD [48] that is a representative RAW-based approach. UltraLED not only performs effective denoising in the dark areas of the scene but also well preserves the details in bright regions, such as the textures of trees, windows, and wall surfaces under light.

## 4.5 Limitations

Although UltraLED significantly outperforms RGB domain approaches in UHDR scenes, it shares a common limitation with RAW denoising methods [48, 54, 21]: the strong dependence of the noise model on camera-specific parameters. We tested the performance of different cameras using their respective noise parameters on our UHDR dataset captured with the SonyA7M4 to demonstrate this more intuitively. The results shown in Tab. 6 indicate the impact of inaccurate noise parameters increases with increasing noise intensity. Consequently, for different camera types, the model needs to be retrained using noise parameters specifically calibrated for the type.

Table 6: Results of the cross-camera experiments.

| Camera Type | ×50 PSNR/SSIM | ×100 PSNR/SSIM | ×200 PSNR/SSIM |
|---|---|---|---|
| NikonD850 | 26.425/0.863 | 25.943/0.825 | 24.784/0.735 |
| SonyA7S2 | 26.333/0.868 | 26.035/0.854 | 25.587/0.796 |
| *SonyA7M4 (matching)* | **27.598/0.898** | **27.327/0.887** | **27.091/0.860** |

## 4.6 Controllable Local Exposure Correction

Since SID [10], most RAW domain denoising methods have achieved varying brightness levels by adjusting the amplification ratios of the input. In contrast, UltraLED applies the ratio primarily to the darkest regions of the image. For areas that are already bright, it adaptively adjusts local exposure, resulting in a more natural appearance. Qualitative results demonstrating this capability are shown in Fig. 6. Notably, with global brightness adjustment alone, as illustrated in the "Input" column, images tend to become overexposed, losing important highlight details. In contrast, our approach effectively preserves fine details and textures while progressively enhancing shadow regions as the amplification ratio increases. This indicates that UltraLED has controllable local exposure correction capability, providing users with a more diverse user experience.

| Input(×1) | Input(×50) | **UltraLED(×50)** | **UltraLED(×100)** | **UltraLED(×200)** |

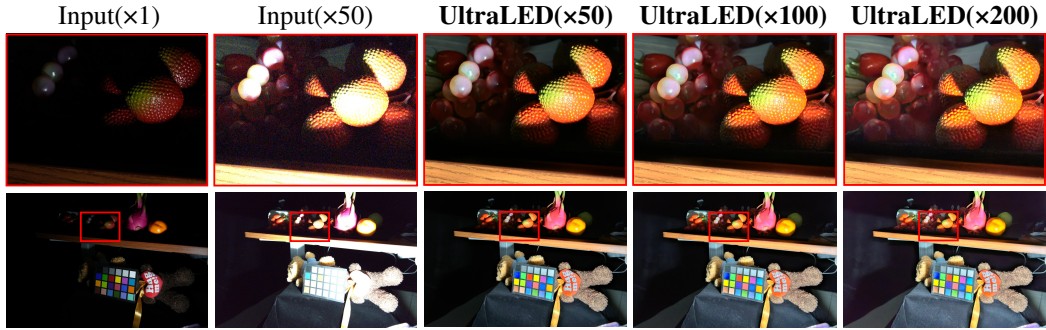

Figure 6: Visualization results on different amplification ratios of the initial RAW image in our model. **Note that these images are all the same input multiplied only by different ratios**. We only show the visualization results for the original input without amplification and the (×50) for the input.

## 5 Conclusion

For UHDR scenes, previous methods typically rely on multi-exposure fusion for image restoration. Single-frame approaches often struggle, either failing to recover details in extremely dark regions or being unable to reconstruct severely overexposed areas. We propose a novel solution that operates in the RAW domain by decoupling exposure correction and denoising. This enables the model to fully leverage the higher bit depth and simpler noise distribution of RAW images. Additionally, we introduce brightness-aware noise modeling and ratio map encoding to guide the recovery of image color and detail. With only a single-frame RAW image, we can see everything in the UHDR scene.

## 6 Acknowledgement

This work was supported in part by the National Natural Science Foundation of China (62306153, 62225604), the Natural Science Foundation of Tianjin, China (25ZXRGGX00290, 24JCJQJC00020, 25JCQNJC01390), the Young Elite Scientists Sponsorship Program by CAST (YESS20240686), the Fundamental Research Funds for the Central Universities (Nankai University, 070-63243143), and Shenzhen Science and Technology Program (JCYJ20240813114237048). The computational devices is partly supported by the Supercomputing Center of Nankai University (NKSC).

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

# Supplementary Material

This supplementary material provides additional insights and results supporting our main paper. We first elaborate on our concept of "see everything" and validate its effectiveness in UHDR scenes (Tab. 7, Fig. 8). We then assess the impact of different denoising networks (Tab. 8) and present detailed visual ablations highlighting the benefits of our RAW domain decoupling strategy (Fig. 10, Fig. 11) and noise modeling (Fig. 13, Fig. 14). Furthermore, we evaluate various ratio map encoding techniques and demonstrate the superiority of our proposed encoding on both the UHDR and ELD datasets [48] (Tab. 9, Tab. 10). Finally, we provide experimental settings, broader societal impact discussions, and license information.

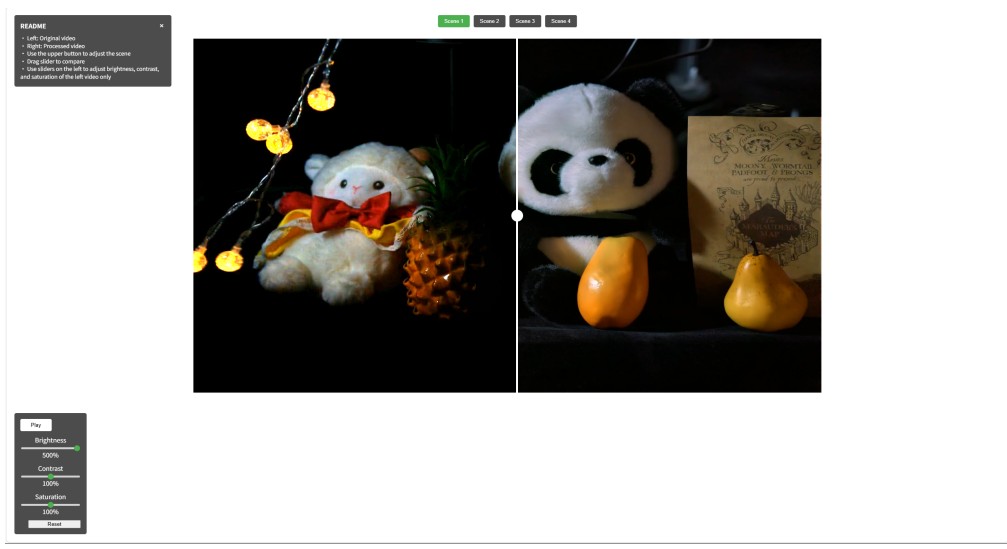

Figure 7: The interface of our demo.

To further demonstrate the effectiveness of UltraLED, **we created a demo included in the supplementary materials**. The interface is shown in Fig. 7. We recorded short videos in UHDR scenes and processed them frame by frame to showcase their performance. These videos simulate real-world scenes where a handheld camera captures scenes with both camera and object motion. Despite the movement, UltraLED can still recover all regions using a short-exposure snapshot, demonstrating its practicality and convenience compared to multi-frame fusion approaches.

## A  More Detailed Explanation of "See Everything"

We define "see everything" as the ability to capture all regions' content in UHDR scenes, including the objects in extremely bright or dark regions, as well as both moving and static elements. A comparison between UltraLED and previous approaches is shown in Tab. 7. Compared with previous methods, UltraLED can "see" all regions in UHDR images.

**Extremely Bright and Extremely Dark Regions.**  Previous single-frame methods struggle with this, particularly those methods [23, 25, 12, 52, 24] based on the RGB domain, which are limited by bit depth and cannot simultaneously preserve rich details in both highlight and shadow areas. RAW-based methods [10, 48, 28, 21, 54], while not restricted by bit depth, often fail to simultaneously recover details across varying brightness levels and perform accurate exposure correction.

**Moving and Static Regions.**  In comparison with the multi-frame HDR fusion techniques [14, 36, 51, 17], UltraLED has a clear advantage in handling motion, especially for moving objects in dark regions. As shown in Fig. 8, in a scene where both the camera and the scene are in motion, even state-of-the-art diffusion-based method [14] fails to recover details in all regions, particularly for moving objects in dark regions. Leveraging the higher potential of RAW images under low-light conditions, our

Table 7: Applicable Regions for Different Methods in UHDR Scenes.

| Domain | Methods | Extremely Bright | | Extremely Dark | |
| --- | --- | --- | --- | --- | --- |
| | | Moving | Static | Moving | Static |
| RGB | Short-exposure-based Methods [55, 23, 25, 2] | ✓ | ✓ | ✗ | ✗ |
| | Long-exposure-based Methods [12, 52, 24, 56] | ✗ | ✗ | ✗ | ✓ |
| | Multi-exposure-based Methods [14, 36, 51, 17] | ✗ | ✓ | ✗ | ✓ |
| RAW | Denoising Methods [10, 48, 28, 21, 54] | ✗ | ✗ | ✓ | ✓ |
| | *UltraLED (Ours)* | ✓ | ✓ | ✓ | ✓ |

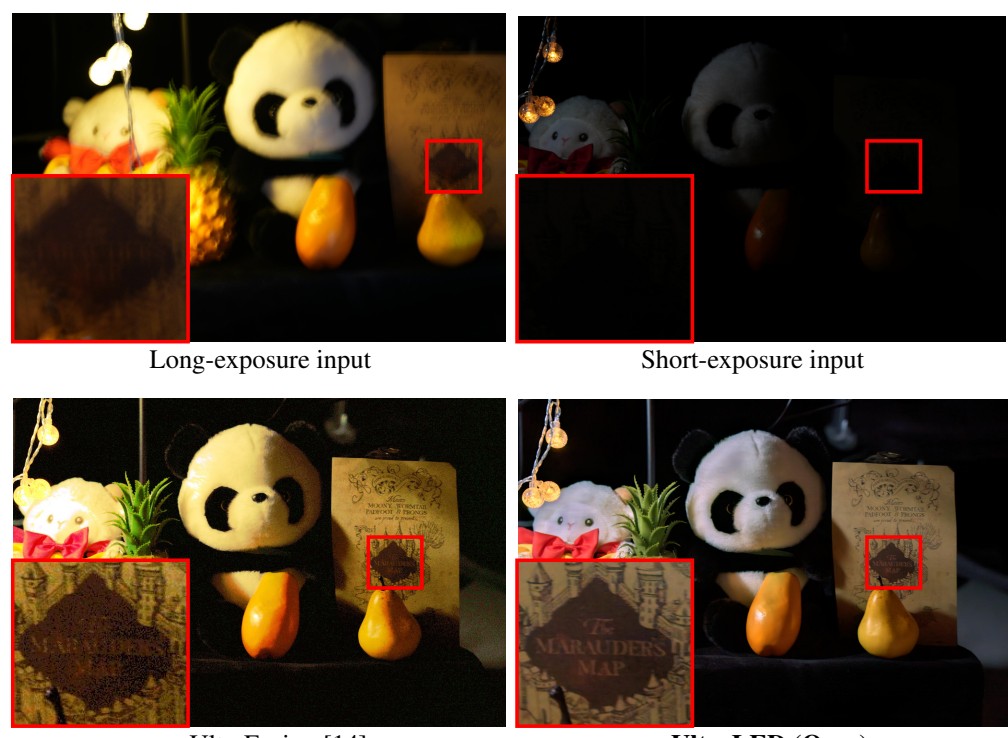

Long-exposure input        Short-exposure input

UltraFusion [14]        **UltraLED (Ours)**

Figure 8: Visual comparison between UltraLED and UltraFusion [14] in a UHDR scene with motion. Note that UltraLED uses a short-exposure single-frame RAW image as input, while UltraFusion takes a short-exposure RGB image and a long-exposure RGB image as inputs. UltraFusion struggles to recover details in dark moving regions and performs poorly in some highlight areas due to alignment difficulties. In contrast, UltraLED, being single-frame and thus alignment-free, successfully restores details in dark regions while avoiding motion artifacts.

approach can recover fine details using only a short exposure. This not only helps capture moving objects without motion blur but also avoids alignment issues, thanks to our single-frame design. These characteristics are especially crucial for dynamic scenes—whether due to camera shake or scene motion—as well as for video applications.

# B  User Study

We conducted a user study to evaluate UltraLED. Specifically, we selected 25 scenes from the UHDR dataset. Since the UHDR dataset mainly consists of indoor scenes, we also captured 11 outdoor UHDR scenes. In addition, we included 15 UHDR scenes from the SID [10] dataset to assess UltraLED's performance on different sensors. A total of 122 participants were invited to take part in the study. For each scene, users were asked to compare the results of our method with those of other baseline methods [48, 2, 23]. The results are shown as Fig. 9, which shows that users consistently

preferred our method over others, indicating that UltraLED produces more visually pleasing images in UHDR scenes. This aligns well with our quantitative evaluation.

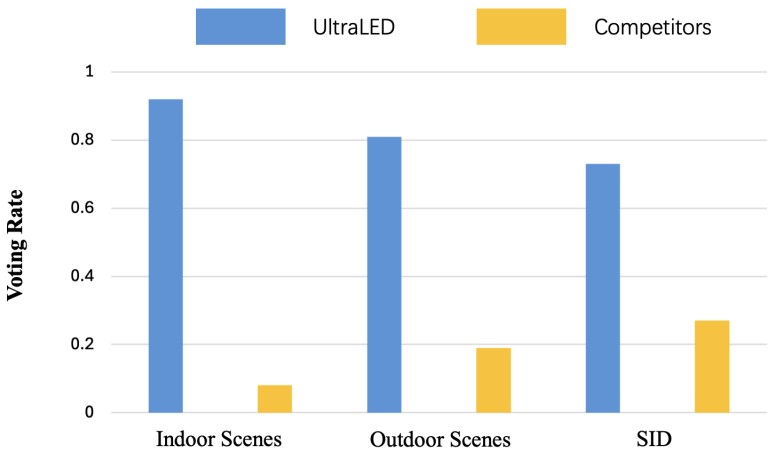

Figure 9: User study on different datasets.

## C   Impact of Different Denoising Networks

We also evaluated the impact of different denoising networks on our results. The results are shown as Tab. 8. These experiments demonstrate that using more advanced network architectures [18, 53] enables the denoising network to better learn the noise model, thereby further enhancing the performance of UltraLED. For a fair comparison, we used U-Net [10] as the denoising network when comparing with other methods.

Table 8: Quantitative comparisons of using different denoising networks as the denoising backbone on the UHDR dataset.

| Network | ×50 PSNR/SSIM | ×100 PSNR/SSIM | ×200 PSNR/SSIM |
|---|---|---|---|
| U-Net [10] | 27.598/0.898 | 27.327/0.887 | 27.091/0.860 |
| ResUnet [18] | 27.741/0.904 | 27.533/0.896 | 27.288/0.882 |
| Restormer [53] | 27.982/0.911 | 27.732/0.903 | 27.437/0.891 |

## D   Visualization Results of Ablation Studies

In this section, we present the visualization results corresponding to the ablation studies in Sec. 4.4.

**RAW Domain Decoupling Strategy.**   The corresponding visualization results are illustrated in Fig. 10 and Fig. 11. Due to the limited bit depth and the complex noise distribution in the RGB domain, RGB-based methods exhibit significantly inferior performance compared to RAW-based methods. This discrepancy is particularly obvious in the recovery of darker regions under extremely high dynamic range scenes. RAW-based methods can reconstruct more realistic details in low-light areas, whereas RGB-based methods suffer from severe color distortion and, in extremely dark regions, often fail to recover meaningful details. In contrast, within the RAW domain, the strategy of decoupling exposure correction and denoising proves markedly beneficial for the restoration of both detail and color in dark regions.

**Brightness-Aware Noise and Ratio Map Encoding.**   The visual results of the ablation study on Brightness-aware Noise and Ratio Map Encoding are shown in Fig. 12. It is evident that both modules contribute positively to noise reduction and detail restoration.

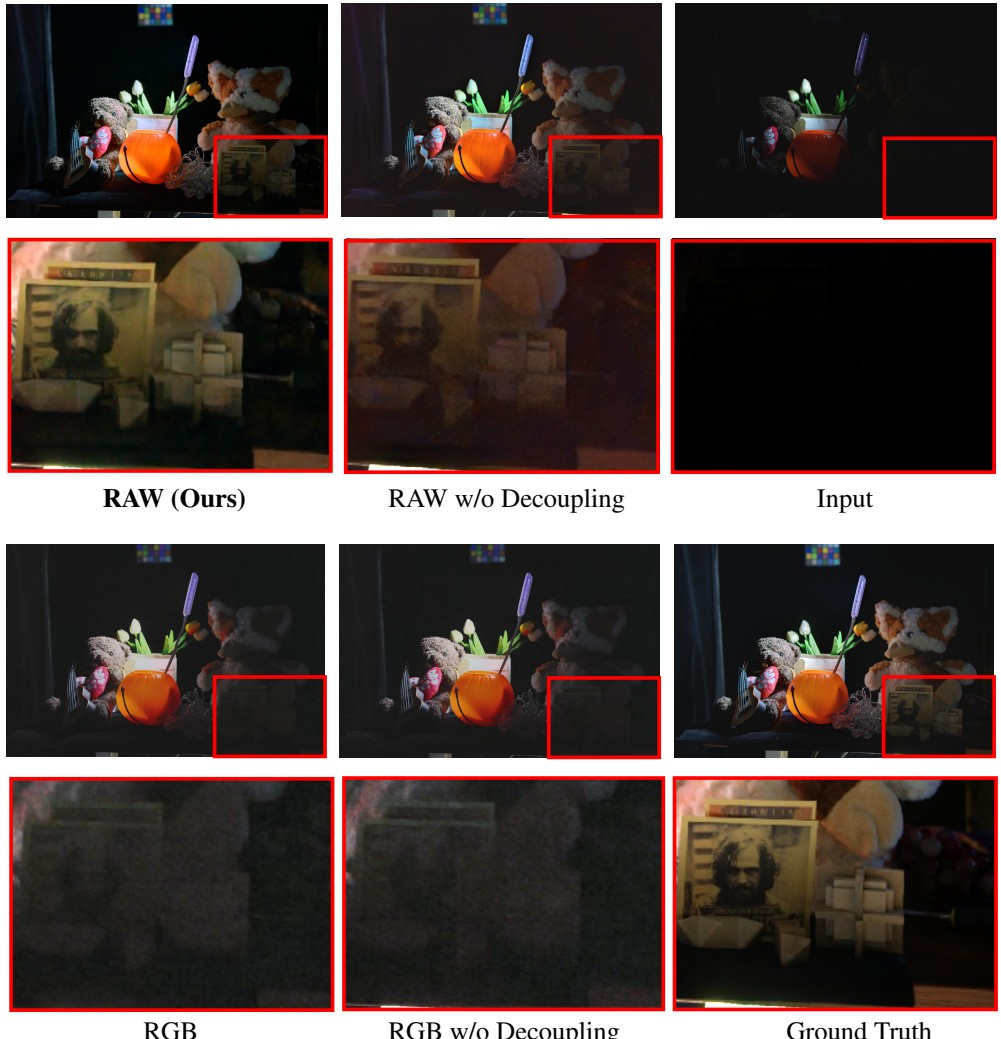

| RAW (Ours) | RAW w/o Decoupling | Input |
|---|---|---|
| RGB | RGB w/o Decoupling | Ground Truth |

Figure 10: Visualization of the ablation experiment results for the RAW domain decoupling strategy. RAW/RGB indicates that the experiment is conducted in the RAW domain/RGB domain, and "w/o Decoupling" means that the decoupling strategy is not used. Pay attention to the details within the red box (Zoomed in and Brightened for Best View). It can be seen that in the RAW domain, the decoupling strategy significantly improves the recovery of details and denoising. The textures of portraits and building blocks are clearer, while in the RGB domain, almost no details can be recovered.

Brightness-Aware Noise is particularly crucial in scenarios with severe noise. Without incorporating this module, the output at ×200 magnification exhibits significant color distortion, as demonstrated in Fig. 13 and Fig. 14.

# E    Ratio Map Encoding

To demonstrate the superiority of our encoding strategy over other methods, we conducted quantitative evaluations using alternative encoding approaches under the same experimental settings, as shown in Tab. 9. The results indicate that our encoding consistently outperforms others across different ratio values. This is because our encoding method is based on a Gaussian distribution, which better captures the continuity of the ratio as well as the correlations between different ratio values. Additionally, the weight represents the overall noise intensity, reinforcing the relationship between the ratio and the noise distribution.

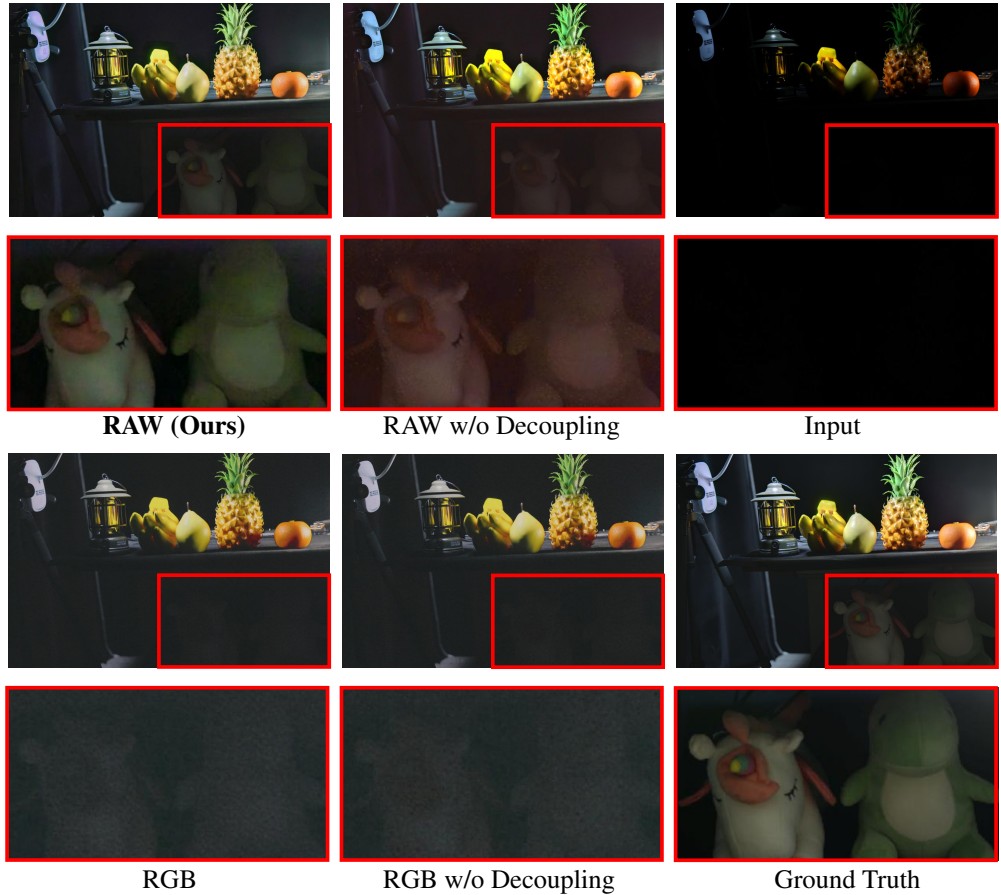

| RAW (Ours) | RAW w/o Decoupling | Input |
| RGB | RGB w/o Decoupling | Ground Truth |

Figure 11: Visualization of the ablation experiment results for the RAW domain decoupling strategy. RAW/RGB indicates that the experiment is conducted in the RAW domain/RGB domain, and "w/o Decoupling" means that the decoupling strategy is not used. Pay attention to the details within the red box (Zoomed in and Brightened for Best View). It can be seen that in the RAW domain, the decoupling strategy significantly improves the recovery of colors and denoising. The colors of the doll are more realistic with less noise, while in the RGB domain, almost no details can be recovered.

Table 9: Quantitative evaluation results of different encoding methods on the UHDR dataset. "None" means no encoding is used. "Onehot" means one-hot encoding is used. and "Position" means position encoding is used. It can be observed that our encoding method provides the most significant performance improvement.

| Encoding | ×50 PSNR/SSIM | ×100 PSNR/SSIM | ×200 PSNR/SSIM |
| --- | --- | --- | --- |
| None | 27.062/0.882 | 26.893/0.871 | 26.734/0.851 |
| Onehot | 27.463/0.893 | 27.201/0.877 | 26.982/0.856 |
| Position | 26.893/0.879 | 27.160/0.872 | 26.781/0.853 |
| *UltraLED (Ours)* | **27.598/0.898** | **27.327/0.887** | **27.091/0.860** |

Additionally, to further showcase the effectiveness and generalization capability of our encoding strategy in guiding denoising under varying brightness levels, we directly tested its impact on denoising performance at different ratios in the RAW domain using the ELD dataset [48]. Since we focused solely on denoising performance and the dynamic range of scenes in the ELD dataset [48] is relatively narrow, each image only requires a single ratio value. It is worth noting that, after normalization, the pixel values of low-light images in the RAW domain are typically smaller than those in the RGB domain—often falling within a range like [0, 0.5]—which tends to result in lower mean squared error and higher PSNR. The quantitative results are shown as Tab. 10. To further

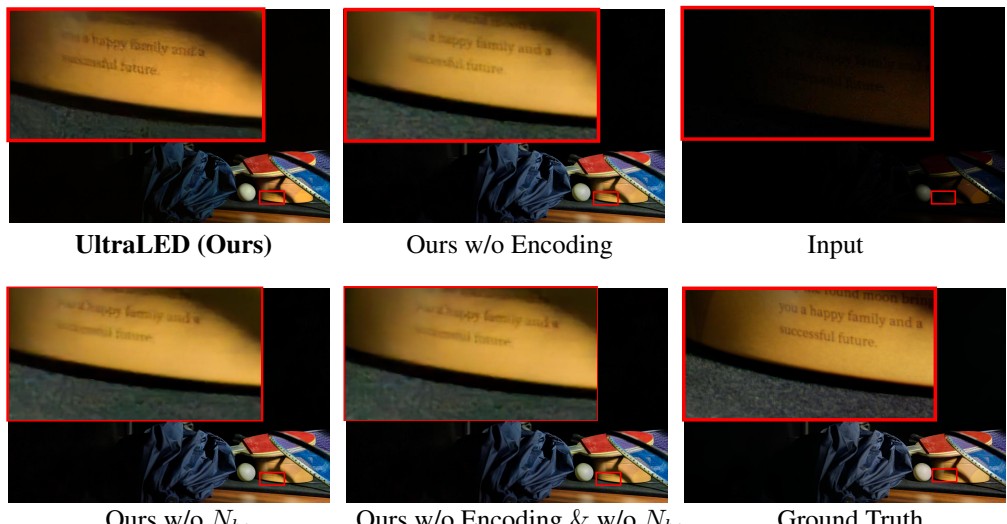

Figure 12: Visualization results of the ablation experiment of Ratio Map Encoding and $N_{ba}$. "w/o Encoding" indicates the case without using Ratio Map Encoding, and "w/o$N_{ba}$" represents the situation without using $N_{ba}$ noise. By observing the changes in the letters, it can be seen that both modules play a positive role in the recovery of details.

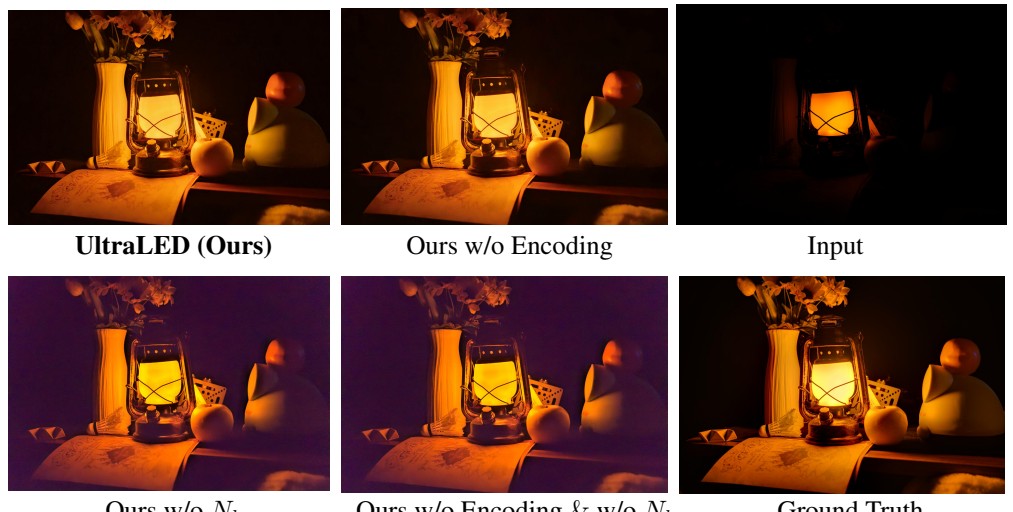

Figure 13: Visualization results of the ablation experiment for Ratio Map Encoding and $N_{ba}$ under strong noise($\times 200$). "w/o Encoding" indicates the case without using Ratio Map Encoding, and "w/o $N_{ba}$" represents the situation without using $N_{ba}$ noise. Pay attention to the differences between the methods with and without using $N_{ba}$ noise. The result of the method without using $N_{ba}$ noise turns obviously purple under strong noise, which seriously affects the visual effect of the image.

demonstrate the generalization capability of our encoding strategy different camera sensors, we evaluate it on two different ELD subsets SonyA7S2 and NikonD850, with ELD noise model. The results are shown as Tab. 11.

## F  Experimental Environments

In our implementations, we used a single NVIDIA GeForce RTX 3090 GPU (24 GB), paired with a 20-core CPU and 64 GB of RAM for training and testing. The Torch version is 1.13.1, and the CUDA version is 12.2. Thanks to the use of only two U-Net architectures, inference on a 7040$\times$4688 RAW image takes approximately 0.85 seconds.

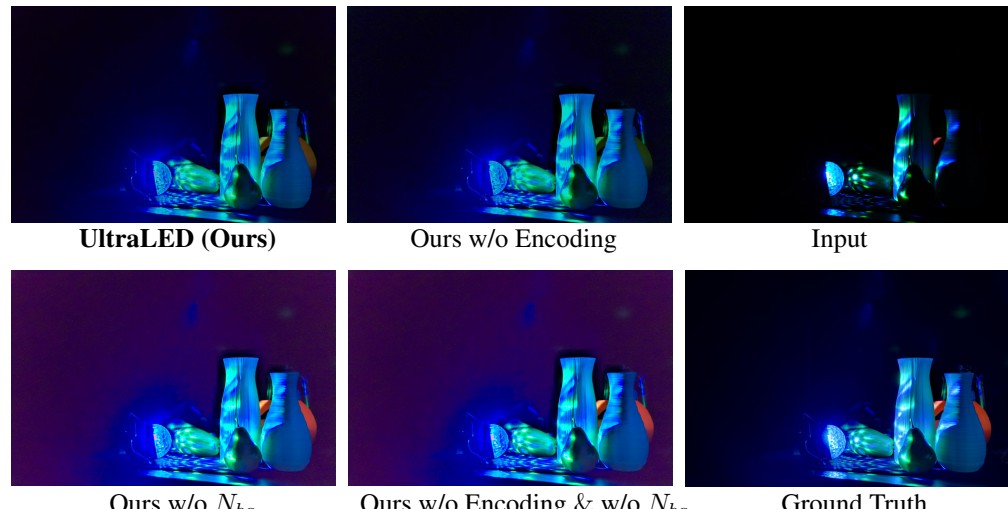

| UltraLED (Ours) | Ours w/o Encoding | Input |
|:---:|:---:|:---:|
| Ours w/o $N_{ba}$ | Ours w/o Encoding & w/o $N_{ba}$ | Ground Truth |

Figure 14: Visualization results of the ablation experiment for Ratio Map Encoding and $N_{ba}$ under strong noise($\times 200$). "w/o Encoding" indicates the case without using Ratio Map Encoding, and "w/o $N_{ba}$" represents the situation without using $N_{ba}$ noise. Pay attention to the differences between the methods with and without using $N_{ba}$ noise. The result of the method without using $N_{ba}$ noise turns obviously purple under strong noise, which seriously affects the visual effect of the image.

Table 10: Quantitative evaluation results of different denoising methods on the ELD [48] dataset. It can be observed that our encoding significantly improves the performance on PG, ELD [48] and SFRN [54], especially under brighter conditions.

| Method | ×1 PSNR/SSIM | ×10 PSNR/SSIM | ×100 PSNR/SSIM | ×200 PSNR/SSIM |
|:---:|:---:|:---:|:---:|:---:|
| PG | 54.974/0.998 | 51.228/0.992 | 42.029/0.885 | 38.059/0.802 |
| *PG+Encoding* | **56.032/0.999** | **51.593/0,993** | **42.132/0.887** | **38.413/0.808** |
| ELD | 53.092/0.997 | 50.940/0.994 | 45.529/0.973 | 43.010/0.947 |
| *ELD+Encoding* | **54.141/0.998** | **51.375/0.995** | **45.646/0.975** | **43.079/0.951** |
| SFRN | 53.478/0.997 | 51.231/0.992 | 45.631/0.977 | 43.013/0.947 |
| *SFRN+Encoding* | **54.662/0.998** | **51.805/0.993** | **45.841/0.980** | **43.021/0.947** |

## G   Broader Impacts

**Potential Positive Impacts.**   UltraLED enhances image quality, enabling users to capture high-quality night scenes without relying on professional equipment. This improvement may also benefit surveillance cameras by providing better performance in low-light conditions, potentially contributing to public safety. Furthermore, it could drive advancements in related industries such as smartphone cameras and security systems, and even support fields like astrophotography and scientific research.

**Potential Negative Impacts.**   While enhanced nighttime surveillance can offer security benefits, it also raises potential privacy concerns. If not properly regulated, the technology might be used in

Table 11: Quantitative evaluation results of different sensors on the ELD [48] dataset.

| Method | ×1 PSNR/SSIM | ×10 PSNR/SSIM | ×100 PSNR/SSIM | ×200 PSNR/SSIM |
|:---:|:---:|:---:|:---:|:---:|
| SonyA7S2 | 53.092/0.997 | 50.940/0.994 | 45.529/0.973 | 43.010/0.947 |
| *SonyA7S2+Encoding* | 54.141/0.998 | 51.375/0.995 | 45.646/0.975 | 43.079/0.951 |
| NikonD850 | 50.673/0.993 | 48.763/0.990 | 43.991/0.966 | 41.821/0.940 |
| *NikonD850+Encoding* | 52.012/0.995 | 49.237/0.991 | 44.178/0.969 | 41.905/0.941 |

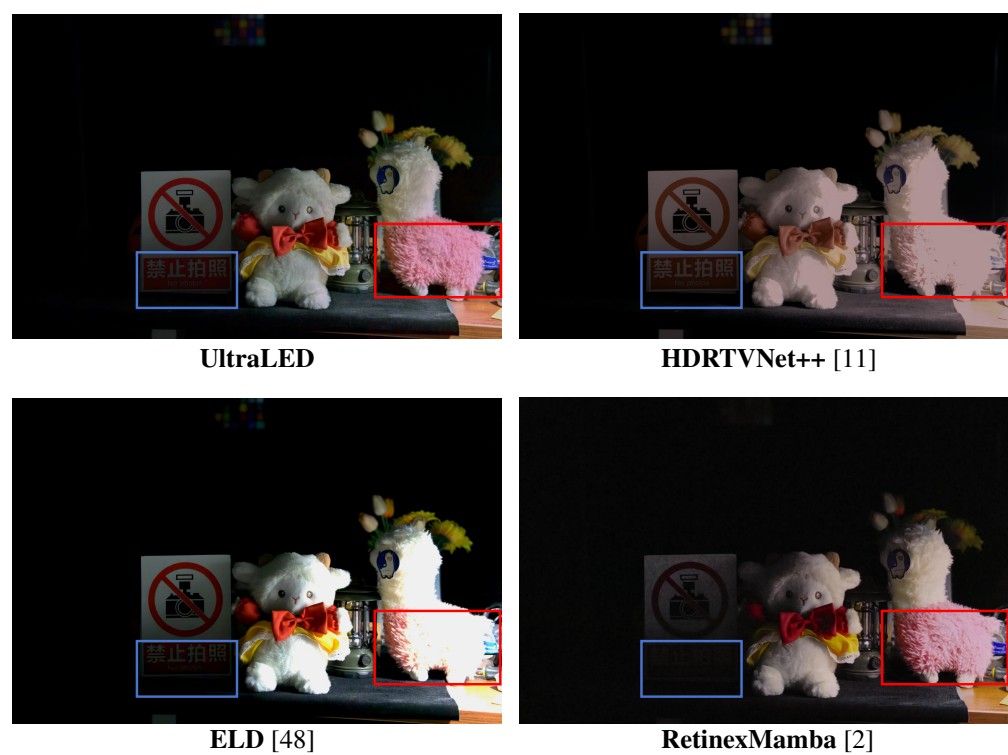

Figure 15: More visualization results on the UHDR dataset.

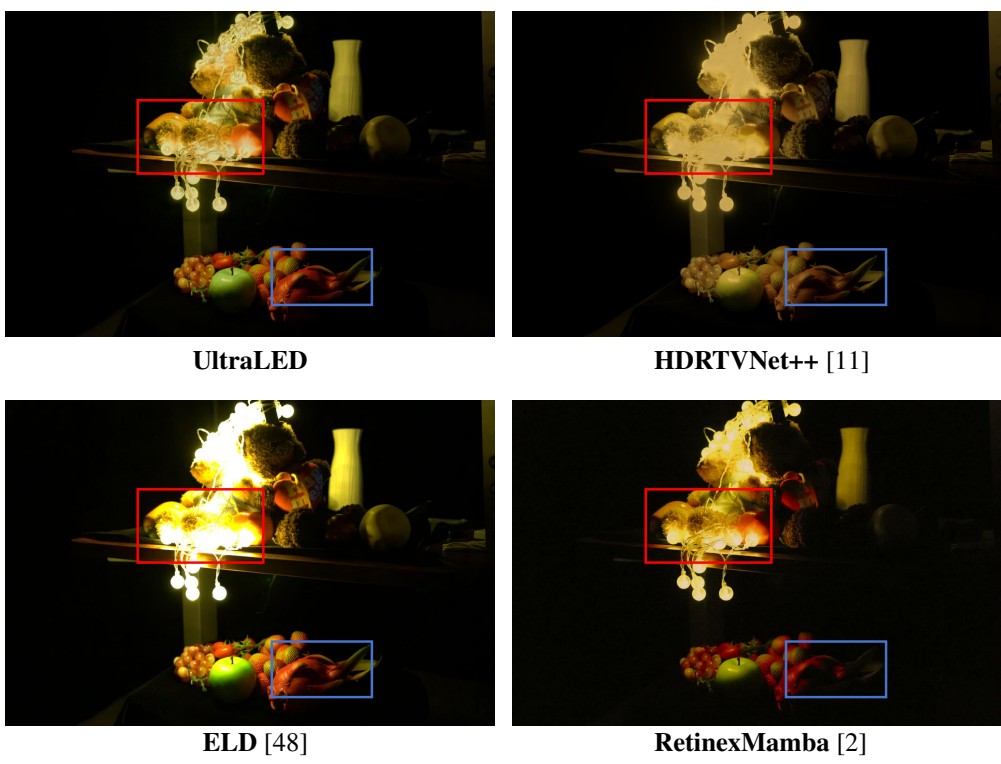

Figure 16: More visualization results on the UHDR dataset.

ways that could unintentionally affect individual privacy, such as in certain forms of unauthorized monitoring or photography.

## H    Licenses for Existing Assets

UltraLED is implemented in the style of BasicSR [47], which is an open source image and video restoration toolbox for super-resolution, denoising, deblurring, etc. We encourage further research based on this codebase. The training data we use is sourced from RawHDR [57]. To synthesize signal-dependent and signal-independent noise in the RAW domain, we adopt the noise model from ELD [48].

## I    More Visualization Results

More visual comparison results are presented in Fig. 15 and Fig. 16. To demonstrate the generality of UltraLED, we also captured and processed images in daytime UHDR scenes. The visual results are shown in Fig. 17.

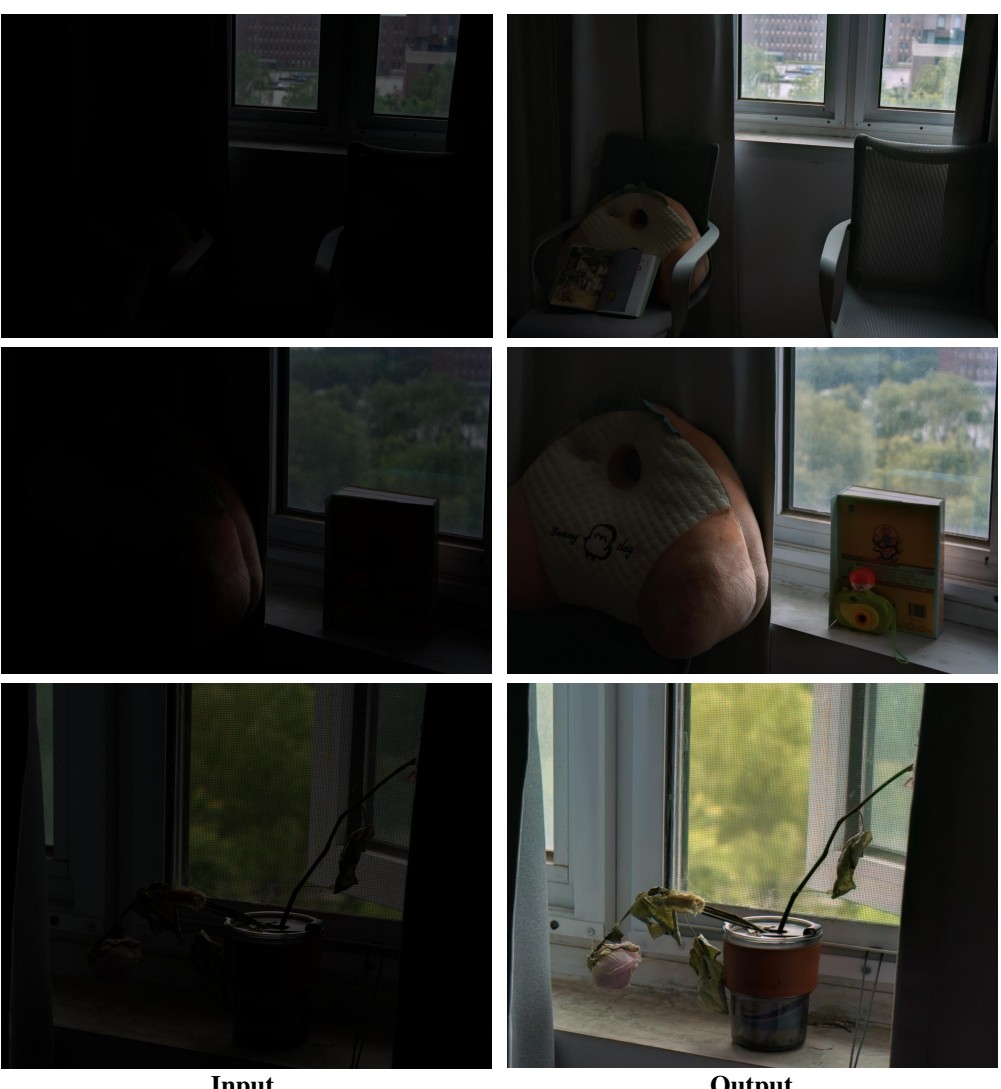

Input                                                                          Output

Figure 17: Visual Results of Processed Daytime Scenes.

