# OpenReview forum: "UltraLED: Learning to See Everything in Ultra-High Dynamic Range Scenes"
_NeurIPS.cc/2025/Conference — NeurIPS 2025 poster_

### Official Review · Reviewer_e9Nm · 2025-06-30

**Clarity:** 2
**Significance:** 2
**Originality:** 2
**Rating:** 4
**Confidence:** 4

**Summary:**

This paper proposes a method for achieving ultra-high dynamic range (UHDR) imaging from a single short-exposure RAW image. The core insight lies in the observation that RAW images possess higher bit depth and more predictable noise characteristics compared to processed images. As a result, even short-exposure RAW captures retain comprehensive scene information, offering substantial potential for low-light image enhancement. The proposed pipeline consists of three main stages: (1) a U-Net architecture is used to predict a ratio map ; (2) this ratio map  guides the generation of a well-exposed image; and (3) a lightweight U-Net is subsequently trained to denoise the generated image.

**Questions:**

1.Please refer to weaknesses.
2.In the comparative analysis, the authors highlight the challenges faced by single-image HDR methods that rely on RGB inputs, implicitly emphasizing the advantages of their single-frame RAW-based approach. However, this comparison may not be entirely fair. RAW images inherently contain significantly more information than RGB images, which are typically post-processed and compressed. As such, it is questionable whether comparing a single RAW image to a single RGB image provides a balanced evaluation. In fact, a single RAW image often has a larger file size and richer content than several RGB frames combined. This raises the question of whether a more appropriate baseline would involve multiple RGB images rather than just one.

**Ethical Concerns:**

["NO or VERY MINOR ethics concerns only"]

**Final Justification:**

The authors' response has addressed most of my concerns, and I decide to raise my final score.

**Limitations:**

Refer to weaknesses and questions, please add relevant explanations.

**Quality:**

2

**Strengths And Weaknesses:**

Strengths：
The author summarizes the problems faced by traditional RGB-based HDR imaging methods and proposes a solution based on RAW image format, providing a new solution to related problems.

Weaknesses：
1.The novelty of this paper is limited. The core components of the proposed method are directly derived from prior work. For example, in Section 3.2, the key component of the method—the ratio map —is adopted from [9] (line 172). In Section 3.1, the strategy for synthesizing artificial overexposure is taken from [38]; the HDR fusion technique is based on [3]; and the noise modeling approach relies on [2].
2.The references, baseline methods, and even the methodological choices are outdated. The authors are encouraged to incorporate more recent studies from the past few years to ensure the timeliness and relevance of their research.
3.The simulation of artificial overexposure in Section 3.1 involves several handcrafted operations, such as adding random highlight patches, using bilateral filtering, and applying Gaussian blur to mimic light diffusion. However, the authors do not provide sufficient details regarding the parameter settings, the underlying rationale for these operations, or any analysis of their impact on the final imaging results.

---

> ### Author Rebuttal · Authors · 2025-07-28
>
> We sincerely thank you for your suggestions. However, we would like to respectfully **clarify several misunderstandings** that may have arisen due to limited familiarity with RAW-domain denoising and noise modeling, and further **elaborate on the key novelty of our method compared to prior works**.
>
> # 1 Clarifications on Misunderstandings
>
> ## 1.1 Ratio Map Is Our Novel Contribution
>
> Contrary to the assumption, **the concept of the *ratio map* does not originate from SID** [1]. Instead, it is an original component of our method, designed for local exposure adjustment and to guide the denoising process. Since SID, most research efforts on short-exposure RAW images have primarily focused on denoising alone (e.g., ELD [2], SFRN [3], PMN [4], LED [5], LLD [6], PNNP [7]). These methods apply a *global (image-wise)* amplification ratio—calculated from the exposure settings (ISO and shutter speed) of the ground-truth and short-exposure images—resulting in the same ratio applied across all pixels.
>
> This leads to overexposure or underexposure in areas with non-uniform brightness distribution, as is evident in many examples in the SID dataset. We illustrate such cases in Figure 4 of our paper. In contrast, our pixel-wise *ratio map* enables **spatially adaptive exposure control**, allowing local modulation of brightness, as shown in Figure 5 of our paper.
>
> ## 1.2 Our Noise Model Is Not Simply Adopted from ELD
>
> Our proposed brightness-aware noise $N_{ba}$ is not a direct application of the ELD noise model. Instead, it is a general implicit noise model that accommodates **any** noise model influenced by exposure adjustment, as theoretically proven in Equation (4). The primary difference between different noise models lies in how the signal-independent noise $N_{in}$ is characterized, taking PG, ELD and SFRN as examples:
>
> - PG assumes a Gaussian distribution,
> - ELD uses a mixture of Gaussian and Tukey-lambda distributions,
> - SFRN leverages empirical distributions sampled from dark frames.
>
> In all cases, our *ratio map encoding* module significantly enhances the model's denoising capability. This is validated in our experiments on the ELD model (see appendix in our supplementary material Table 4). Additional results on the ELD SonyA7S2 subset using both PG and SFRN noise models are shown below:
>
> |               |        ×1        |       ×10        |       ×100       |       ×200       |
> | :-----------: | :--------------: | :--------------: | :--------------: | :--------------: |
> |               |    PSNR/SSIM     |    PSNR/SSIM     |    PSNR/SSIM     |    PSNR/SSIM     |
> |      PG       |   54.974/0.998   |   51.228/0.992   |   42.029/0.885   |   38.059/0.802   |
> |  PG + Encoding  | **56.032/0.999** | **51.593/0,993** | **42.132/0.887** | **38.413/0.808** |
> |     SFRN      |   53.478/0.997   |   51.231/0.992   |   45.631/0.977   |   43.013/0.947   |
> | SFRN + Encoding | **54.662/0.998** | **51.805/0.993** | **45.841/0.980** | **43.021/0.947** |
>
> **We emphasize the ELD model because it is widely adopted in both academia and industry, it can be easily replaced by any noise models.** Our framework is flexible and agnostic to the specific noise formulation.
>
> ## 1.3 Realism of Synthetic Overexposure
>
> Beyond using Flare7K [8] to simulate overexposure, we introduced an additional strong degradation model that accounts for the complexity of brightness irregularities in low-light environments. In such scenes, intensity variation is often not caused by direct light sources but by complex interactions such as reflection and scattering. As mentioned in works like *IClight* [9], given the impracticality of precisely modeling all degradations, we adopt a broader model to better capture real-world variability. And its parameter settings will be released along with the code.
>
>
>
> # 2 On the Question of Novelty
>
> We respectfully disagree with the assessment that our method lacks novelty. To the best of our knowledge, **this is the first work to reconstruct HDR images from short-exposure RAW inputs**, and more importantly, the **first to simultaneously address both denoising and HDR tone mapping** for such inputs. This is a challenge explicitly raised by the authors of the SID [1] in their *discussion* section, yet it has remained unsolved for years.
>
> Previous studies have mainly focused on improving the precision or usability of noise modeling (e.g., ELD [2], SFRN [3], PMN [4], LED [5], LLD [6], PNNP [7]), without addressing the joint exposure correction and denoising problem. Our method fills this gap, as evidenced by Figure 4 in our paper.
>
> The practical significance of this contribution is considerable: for UHDR scenes with camera shake or object motion, acquiring multiple frames is often infeasible. Our method enables accurate HDR reconstruction from a **single short-exposure RAW frame**, achieving a capability that no existing approach has demonstrated.
>
> **We believe it is unfair to claim our work lacks novelty when it successfully solves a long-standing, well-recognized problem in the field**.
>
> Even when compared to state-of-the-art methods such as *RetinexMamba* [10] or other representative baselines, our method shows clear advantages in challenging UHDR conditions.
>
>
>
> # 3 On Fairness of Comparisons
>
> **We have conducted fair comparisons with RAW-based denoising** [2, 5] **followed by image reconstruction**, which is the most reasonable strategy for UHDR recovery from single-frame RAW input for previous methods (the way they are synthesised after denoising is the same as our ground truth). Our proposed $N_{ba}$ and *ratio map encoding* modules enable us to outperform such baselines within the RAW domain itself, as shown in Table 2 of our paper.
>
> Regarding the concern that single-frame RAW is not meaningful unless compared to multi-frame RGB methods, we respectfully disagree. **There are many real-world scenarios where multi-frame capture is not possible**, such as flash photography or fast motion. This is demonstrated in our supplementary demo scenes, where multi-frame methods often fail due to motion artifacts.
>
> Moreover, every RGB image is ultimately derived from a RAW image. Thus, **RAW-based single-frame recovery is not only meaningful but sometimes the only viable solution**. Our work is aimed precisely at solving this class of problems.
>
> Despite this, we still performed a direct comparison through a user study with *UltraFusion*, a state-of-the-art multi-frame RGB-based HDR method. We captured 18 real-world UHDR scenes (mainly urban night scenes). Our method only used RAW images captured at ISO 200, 1/400s shutter, while UltraFusion fused two RGB frames: the same short exposure and an additional EV0 exposure.
>
> All data was captured handheld, introducing mild camera shake and natural motion (e.g., pedestrians, vehicles). A total of 122 users participated in the evaluation. Additionally, we also included comparisons with ELD, another single-frame RAW method. The results indicate that 83% of our results were preferred. The results are as follows:
>
> | Method | ELD  | UltraFusion | Ours |
> | :----: | :--: | :---------: | :--: |
> | Winner |  6%  |     11%     | 83%  |
>
> In regions affected by camera motion or object movement, where alignment across frames becomes unreliable, UltraFusion often produces results with noticeable texture loss or unrealistic details. In contrast, our method is capable of faithfully reconstructing fine details and preserving complete textures. **This advantage is clearly illustrated in the comparison with UltraFusion shown in Figure 2 of the appendix in our supplementary material**. Furthermore, in cases where images at EV0 exhibit significant motion blur, our method—relying solely on short-exposure inputs—is still able to recover sharp representations of moving objects. UltraFusion, on the other hand, tends to retain motion blur artifacts in these regions due to its reliance on long-exposure frames.
>
> **We will make the RAW images and results from our user study publicly available upon the paper's review results to ensure transparency and reproducibility**.
>
>
> [1] Learning to See in the Dark
>
> [2] Physics-based Noise Modeling for Extreme Low-light Photography
>
> [3] Rethinking Noise Synthesis and Modeling in Raw Denoising
>
> [4] Learnability Enhancement for Low-light Raw Denoising: Where Paired Real Data Meets Noise Modeling
>
> [5] Lighting Every Darkness in Two Pairs: A Calibration-Free Pipeline for RAW Denoising
>
> [6] Physics-Guided ISO-Dependent Sensor Noise Modeling for Extreme Low-Light Photography
>
> [7] Physics-guided Noise Neural Proxy for Practical Low-light Raw Image Denoising
>
> [8] Flare7K: A Phenomenological Nighttime Flare Removal Dataset
>
> [9] Scaling In-the-Wild Training for Diffusion-based Illumination Harmonization and Editing by Imposing Consistent Light Transport
>
> [10] Retinexmamba: Retinex-based Mamba for Low-light Image Enhancement

---

> > ### Comment · Reviewer_e9Nm · 2025-08-08
> >
> > Thank you for the authors' response. I have carefully read the rebuttal, and most of my concerns have been addressed. I will raise my score.

---

> > > ### Author Response · Authors · 2025-08-08
> > >
> > > We sincerely appreciate your recognition. Your valuable suggestions will be carefully incorporated to further enhance the quality of our work. It has been a pleasure to engage in this discussion.

---

> ### Author Response · Authors · 2025-08-06
>
> Dear Reviewer e9Nm,
>
> Thank you for your time and review.
>
> Does our response solve your concerns? If you have any further comments or require additional clarification, we would greatly appreciate your continued feedback.
>
> Best regards,
>
> The authors

---

> ### Author Response · Authors · 2025-08-08
>
> Dear Reviewer e9Nm,
>
> I hope this message finds you well. As there is only one day remaining before the discussion period concludes, we would like to express our sincere gratitude for your valuable suggestions. We have provided targeted responses to address your comments and would greatly appreciate it if you could review them to see whether they resolve your concerns.
>
> If there are any remaining issues, please do let us know. Your insights are of great importance to us, and we would be glad to engage in further discussion or provide additional clarification.
>
> Thank you for your time and thoughtful feedback.
>
> Best regards,
>
> The Authors

---

### Official Review · Reviewer_UcfZ · 2025-07-01

**Clarity:** 1
**Significance:** 2
**Originality:** 2
**Rating:** 4
**Confidence:** 2

**Summary:**

This paper presents a method for reconstructing ultra-high dynamic range (UHDR) images from a single short-exposure RAW image. The proposed two-stage framework performs exposure correction and brightness-aware denoising to recover details in dark regions. A custom dataset with 9-stop bracketing is used for training, and experiments show significant improvements over existing single-frame methods.

**Questions:**

Is this the first paper to address HDR reconstruction from a single short-exposure RAW image?

In Section 4.5, the method shows good performance only on specific camera types, but fails to generalize to others, which is a significant limitation. Could the authors provide some experimental results to demonstrate how poorly the model performs on different types of cameras without retraining?

**Ethical Concerns:**

["NO or VERY MINOR ethics concerns only"]

**Final Justification:**

The authors' rebuttal has addressed my concerns. Therefore, I have decided to raise my rating.

**Limitations:**

yes

**Quality:**

2

**Strengths And Weaknesses:**

The proposed method achieves excellent performance, as shown in Tables 1, 2, and 3, outperforming previous methods such as HDRTVNet, RetinexMamba, and others.

In addition, the authors also build a corresponding dataset for benchmarking UHDR reconstruction performance. However, the synthesized UHDR images lack thorough validation, making it uncertain whether they can accurately evaluate the model's performance.

Moreover, the writing quality of the paper is quite poor, and it's difficult to clearly understand the contributions. Is the paper merely proposing a data synthesis pipeline? The network architecture seems to be entirely based on previous methods.

As mentioned in Section 4.5, the strong dependence of the noise model on camera-specific parameters makes it difficult to apply the method in real-world scenarios.

---

> ### Author Rebuttal · Authors · 2025-07-27
>
> We are sincerely grateful for your valuable comments and suggestions.
> # 1 Novelty and Practical Relevance
>
> **Yes,** **our work is the first to leverage a short-exposure RAW image for HDR reconstruction.** Also, **it is the first to jointly address the challenges of denoising and HDR tone mapping from short-exposure RAW input**—a long-standing issue highlighted by SID [1] but left unsolved for years. In the discussion section of the original SID paper, the authors noted the difficulty of simultaneously removing noise and performing HDR tone mapping from short-exposure RAW data.
>
> Subsequent research in this domain has largely focused on developing more accurate or more accessible noise modeling techniques (e.g., ELD [2], SFRN [3], PMN [4], LED[5], LLD[6], PNNP[7]). However, our method is the first to effectively tackle this dual challenge, as shown in Figure 4 of our paper. This advancement holds strong practical significance: in UHDR scenes, it is common to encounter camera shake or dynamic elements (e.g., moving objects). Our method enables high-fidelity HDR reconstruction even from extremely short exposures—something that no prior approach has been able to achieve.
>
>
>
> # 2 Experiments using non-matching cameras without retraining
>
> The concern you raised likely stems from a misunderstanding of RAW-domain processing. In practice, each camera exhibits a unique noise distribution in the RAW domain. Fortunately, **many well-established techniques** [2-7] **exist to estimate a camera’s noise model and parameters**, and they are capable of producing reliable results. Consequently, it is rare—both in academia and industry—for RAW image processing to be conducted using inaccurate noise models without retraining.
>
> Nevertheless, in response to your question, we present cross-camera experiments on our dataset (captured with SonyA7M4). In these experiments, **we applied noise parameters from non-matching camera models to test our dataset**:
>
> |  Camera Type |     ×50      |     ×100     |     ×200     |
> | :-------: | :----------: | :----------: | :----------: |
> |           |  PSNR/SSIM   |  PSNR/SSIM   |  PSNR/SSIM   |
> | NikonD850 | 26.425/0.863 | 25.943/0.825 | 24.784/0.735 |
> | SonyA7S2  | 26.333/0.868 | 26.035/0.854 | 25.587/0.796 |
> | SonyA7M4 (matching)  | **27.598/0.898** | **27.327/0.887** | **27.091/0.860** |
>
> It can be seen that the impact of inaccurate noise parameters increases with increasing noise intensity.
>
>
> # 3 Generalization to Different Cameras
>
> Based on the derivation in Equation (4) in our paper, **our brightness-aware noise $N_{ba}$ can adapt well to any camera-specific noise model**. The variation across noise models primarily lies in different modeling of the signal-independent noise $N_{in}$. For example, PG assumes a Gaussian distribution; ELD [2] uses a mixture of Gaussian and Tukey-lambda distributions; and SFRN [3] relies on real dark frame sampling.
>
> **Our ratio map encoding module consistently enhances denoising performance across various types of camera sensors and noise models**.
>
> We have already provided thorough evidence of its effectiveness on the SonyA7M4 sensor within our paper. To further demonstrate the generalization capability of our method across different camera sensors, we evaluate it on two additional ELD subsets—SonyA7S2 and NikonD850—with ELD noise model:
>
> |                      |        ×1        |       ×10        |       ×100       |       ×200       |
> | :------------------: | :--------------: | :--------------: | :--------------: | :--------------: |
> |                      |    PSNR/SSIM     |    PSNR/SSIM     |    PSNR/SSIM     |    PSNR/SSIM     |
> |       SonyA7S2       |   53.092/0.997   |   50.940/0.994   |   45.529/0.973   |   43.010/0.947   |
> | SonyA7S2 + Encoding  | **54.141/0.998** | **51.375/0.995** | **45.646/0.975** | **43.079/0.951** |
> |      NikonD850       |   50.673/0.993   |   48.763/0.990   |   43.991/0.966   |   41.821/0.940   |
> | NikonD850 + Encoding | **52.012/0.995** | **49.237/0.991** | **44.178/0.969** | **41.905/0.941** |
>
> In addition, to validate the robustness of our method under different noise modeling assumptions, we conduct experiments on the Sony subset of the ELD dataset using three distinct noise models: a simple parametric Gaussian model (PG), a physically calibrated noise model (ELD), and a data-driven real-world noise model (SFRN). The results further confirm the adaptability and generalization strength of our approach:
>
> |                 |        ×1        |       ×10        |       ×100       |       ×200       |
> | :-------------: | :--------------: | :--------------: | :--------------: | :--------------: |
> |                 |    PSNR/SSIM     |    PSNR/SSIM     |    PSNR/SSIM     |    PSNR/SSIM     |
> |       PG        |   54.974/0.998   |   51.228/0.992   |   42.029/0.885   |   38.059/0.802   |
> |  PG + Encoding  | **56.032/0.999** | **51.593/0,993** | **42.132/0.887** | **38.413/0.808** |
> |       ELD       |   53.092/0.997   |   50.940/0.994   |   45.529/0.973   |   43.010/0.947   |
> | ELD + Encoding  | **54.141/0.998** | **51.375/0.995** | **45.646/0.975** | **43.079/0.951** |
> |      SFRN       |   53.478/0.997   |   51.231/0.992   |   45.631/0.977   |   43.013/0.947   |
> | SFRN + Encoding | **54.662/0.998** | **51.805/0.993** | **45.841/0.980** | **43.021/0.947** |
>
> These results clearly demonstrate that our framework is robust and generalizable to different camera sensors and noise models.
>
>
>
> [1] Learning to See in the Dark
>
> [2] Physics-based Noise Modeling for Extreme Low-light Photography
>
> [3] Rethinking Noise Synthesis and Modeling in Raw Denoising
>
> [4] Learnability Enhancement for Low-light Raw Denoising: Where Paired Real Data Meets Noise Modeling
>
> [5] Lighting Every Darkness in Two Pairs: A Calibration-Free Pipeline for RAW Denoising
>
> [6] Physics-Guided ISO-Dependent Sensor Noise Modeling for Extreme Low-Light Photography
>
> [7]  Physics-guided Noise Neural Proxy for Practical Low-light Raw Image Denoising

---

> ### Author Response · Authors · 2025-08-06
>
> Dear Reviewer UcfZ,
>
> Thank you for your time and review.
>
> Does our response solve your concerns? If you have any further comments or require additional clarification, we would greatly appreciate your continued feedback.
>
> Best regards,
>
> The authors

---

> > ### Comment · Reviewer_UcfZ · 2025-08-06
> >
> > Thank you for your response, which has addressed my concerns. Therefore, I have decided to raise my rating. The experiment on generalization to different cameras is crucial for demonstrating the effectiveness of the proposed method. I suggest the authors include this in the final version of the paper.

---

> > > ### Author Response · Authors · 2025-08-06
> > >
> > > Thank you for your kind recognition. We will incorporate the response into the final version and further enhance the quality of the paper. We appreciate the opportunity to engage in this valuable discussion.

---

### Official Review · Reviewer_2iPX · 2025-07-01

**Clarity:** 4
**Significance:** 4
**Originality:** 4
**Rating:** 6
**Confidence:** 3

**Summary:**

This papers presents a new single-shot method for capturing UHDR content. The method relies on heavily underexposed RAW images, which are then adaptively exposed based on new learned ratio maps, and brightness-aware denoising. The results are of very high quality, translating naturally to video as well. The networks used are simple and efficient, making the method highly practical. The quantitative and qualitative comparisons clearly show the superiority of the method, and the ablations demonstrate the effectiveness of the whole approach.

**Questions:**

Figure 3 in the paper shows more colorful results than the ground truth, which could be in principle more desirable, but at the same time it can introduce some inconsistencies. For example, in the last row, the ground truth shows an evenly desaturated green mouth region, while the produced result has a more uneven distribution. Is this coming from the use of standard isp and exposure fusion for the tone mapping? Could there be some extra steps to mitigate it?

**Ethical Concerns:**

["NO or VERY MINOR ethics concerns only"]

**Final Justification:**

The authors replied to my minor comments. I have read the other reviews and discussion, and I’m still keeping my original score. I believe the quality and elegance of the solution is worth acceptinng.

**Limitations:**

yes

**Quality:**

4

**Strengths And Weaknesses:**

- Strengths:
- A very practical method that checks all the boxes compared with previous works.
- Simple and effective 2-step approach that makes the problem very tractable.
- Very high quality results, way ahead of previous works.
- Clear presentation of problem, context, method and evaluations.
- A carefully created UHDR dataset that may help future research in this space.

- Weaknesses:
- As the authors acknowledge, different cameras may have different noise profiles, which would require retraining the noise models. However, it is would be very feasible for different vendors to train those modules themselves.

---

> ### Author Rebuttal · Authors · 2025-07-27
>
> Sincerely thank you for your recognition of our work.
>
> # 1 About Inconsistent Tones
>
> Yes, the difference you mentioned primarily arise from the inverse ISP process employed during exposure fusion for synthetic data generation. The tone and style of an image are closely related to the specific preferences of ISP adopted by different vendors. For manufacturers, it is relatively straightforward to adjust the tonal rendering to match their desired aesthetic. As a RAW-to-RAW method, our goal is to demonstrate the feasibility and effectiveness of our approach, and thus we intentionally employ a minimal and generic ISP pipeline without incorporating any vendor-specific tuning or stylistic enhancement.
>
>
>
> # 2 Generalization to Different Cameras
>
> Regarding the weakness, while it is true that our model requires retraining for different sensor types, the cost of retraining for different manufacturers is relatively low. Moreover, as derived in Equation (4), **our brightness-aware noise $N_{ba}$ is designed to adapt well to any noise model associated with any camera**. The primary variation between noise models lies in the modeling of the signal-independent noise $N_{in}$: for instance, the PG model uses a Gaussian distribution, the ELD [4] model combines Gaussian and Tukey-lambda mixture distributions, and the SFRN [5] model samples real noise from dark frames.
>
> Importantly, **our ratio map encoding module consistently enhances denoising performance across various types of camera sensors and noise models**.
>
> We have already provided thorough evidence of its effectiveness on the SonyA7M4 sensor within our paper. To further demonstrate the generalization capability of our method across different camera sensors, we evaluate it on two additional ELD subsets—SonyA7S2 and NikonD850—with ELD noise model:
>
> |                      |        ×1        |       ×10        |       ×100       |       ×200       |
> | :------------------: | :--------------: | :--------------: | :--------------: | :--------------: |
> |                      |    PSNR/SSIM     |    PSNR/SSIM     |    PSNR/SSIM     |    PSNR/SSIM     |
> |       SonyA7S2       |   53.092/0.997   |   50.940/0.994   |   45.529/0.973   |   43.010/0.947   |
> | SonyA7S2 + Encoding  | **54.141/0.998** | **51.375/0.995** | **45.646/0.975** | **43.079/0.951** |
> |      NikonD850       |   50.673/0.993   |   48.763/0.990   |   43.991/0.966   |   41.821/0.940   |
> | NikonD850 + Encoding | **52.012/0.995** | **49.237/0.991** | **44.178/0.969** | **41.905/0.941** |
>
> In addition, to validate the robustness of our method under different noise modeling assumptions, we conduct experiments on the Sony subset of the ELD dataset using three distinct noise models: a simple parametric Gaussian model (PG), a physically calibrated noise model (ELD), and a data-driven real-world noise model (SFRN). The results further confirm the adaptability and generalization strength of our approach:
>
> |                 |        ×1        |       ×10        |       ×100       |       ×200       |
> | :-------------: | :--------------: | :--------------: | :--------------: | :--------------: |
> |                 |    PSNR/SSIM     |    PSNR/SSIM     |    PSNR/SSIM     |    PSNR/SSIM     |
> |       PG        |   54.974/0.998   |   51.228/0.992   |   42.029/0.885   |   38.059/0.802   |
> |  PG + Encoding  | **56.032/0.999** | **51.593/0,993** | **42.132/0.887** | **38.413/0.808** |
> |       ELD       |   53.092/0.997   |   50.940/0.994   |   45.529/0.973   |   43.010/0.947   |
> | ELD + Encoding  | **54.141/0.998** | **51.375/0.995** | **45.646/0.975** | **43.079/0.951** |
> |      SFRN       |   53.478/0.997   |   51.231/0.992   |   45.631/0.977   |   43.013/0.947   |
> | SFRN + Encoding | **54.662/0.998** | **51.805/0.993** | **45.841/0.980** | **43.021/0.947** |
>
> These results clearly demonstrate that our framework is robust and generalizable to different camera sensors and noise models.
>
>
>
> [1] Beyond learned metadata-based raw image reconstruction
>
> [2] Unprocessing Images for Learned Raw Denoising
>
> [3] Metadata based raw reconstruction via implicit neural functions
>
> [4] Physics-based Noise Modeling for Extreme Low-light Photography
>
> [5] Rethinking Noise Synthesis and Modeling in Raw Denoising

---

> > ### Comment · Reviewer_2iPX · 2025-08-04
> >
> > Thank you for your replies. I don't have any other questions

---

### Official Review · Reviewer_N566 · 2025-07-02

**Clarity:** 2
**Significance:** 3
**Originality:** 3
**Rating:** 3
**Confidence:** 4

**Summary:**

This paper studies reconstruction of ultra-high dynamic range scenes from a single short-exposure RAW image. It proposes a two-stage pipeline—first predicting a pixel-wise ratio map for adaptive exposure correction, then applying a brightness-aware noise model to guide denoising—fully leveraging RAW’s high bit depth. Contributions include (1) decoupling exposure and noise restoration, (2) a novel data synthesis workflow combining simulated over-exposure, HDR fusion, and realistic noise modeling, and (3) a purpose-built UHDR dataset.

**Questions:**

Could you include experiments on scenes with motion (e.g., moving objects or camera motion) and compare against a state-of-the-art multi-frame fusion method such as UltraFusion?

**Ethical Concerns:**

["NO or VERY MINOR ethics concerns only"]

**Quality:**

3

**Strengths And Weaknesses:**

Strengths

- The paper is technically sound, with a clear problem formulation and well-justified two-stage design that decouples exposure correction from denoising.

- The motivation for single-frame RAW-domain UHDR reconstruction is crisply presented: avoiding misalignment and ghosting in dynamic scenes.

- Demonstrates that a single short-exposure RAW frame suffices to recover both highlights and deep shadows, a capability critical for real-world night and mixed-lighting scenarios.

Weaknesses

- The core concern is that the paper’s main claim—single-frame RAW reconstruction avoids the ghosting artifacts of multi-frame exposure fusion in dynamic scenes—can only be validated by directly comparing against strong multi-frame fusion baselines (e.g., UltraFusion). Without such head-to-head experiments, the asserted advantage of using a single frame remains unproven and the contribution loses its significance.

- The use of two UNets and multi-step data synthesis may incur higher runtime or memory requirements than existing single-stage RGB-domain approaches; a detailed analysis of inference speed and model footprint is missing.

[1] UltraFusion: Ultra High Dynamic Imaging using Exposure Fusion

---

> ### Author Rebuttal · Authors · 2025-07-28
>
> We sincerely appreciate your valuable comments and suggestions.
>
> # 1 **Comparison with Multi-frame Method**
>
> Since the ground truth for multi-frame methods and single-frame methods is not easily unified, **we conducted a user study head-to-head comparing our approach with UltraFusion** [1]. Specifically, we captured 18 scenes from everyday environments, primarily urban night scenes under ultra-high dynamic range (UHDR) conditions. Our method utilized only a single short-exposure RAW image captured at ISO 200 and 1/400s shutter speed, while UltraFusion leveraged both an RGB image with the same exposure and an additional EV0 RGB image for fusion.
>
> All scenes were captured handheld, introducing natural camera shake, and some included dynamic elements such as moving pedestrians or vehicles. In addition, this dataset was also used to compare our results with ELD [2], another single-frame RAW-based method. A total of 122 participants evaluated the results, and 83% of the images processed by our method were preferred. The following table summarizes the percentage of votes each method received:
>
> | Method | ELD  | UltraFusion | Ours |
> | :----: | :--: | :---------: | :--: |
> | Winner |  6%  |     11%     | 83%  |
>
> In regions affected by camera motion or object movement, where alignment across frames becomes unreliable, UltraFusion often produces results with noticeable texture loss or unrealistic details. In contrast, our method is capable of faithfully reconstructing fine details and preserving complete textures. **This advantage is clearly illustrated in the comparison with UltraFusion shown in Figure 2 of the appendix in our supplementary material**. Furthermore, in cases where images at EV0 exhibit significant motion blur, our method—relying solely on short-exposure inputs—is still able to recover sharp representations of moving objects. UltraFusion, on the other hand, tends to retain motion blur artifacts in these regions due to its reliance on long-exposure frames.
>
> **We will make the RAW images and results from our user study publicly available upon the paper's review results to ensure transparency and reproducibility**.
>
>
> # 2 **Inference Speed and Model Memory**
>
> Regarding the concern over computational and memory efficiency, our approach benefits from a lightweight, decoupled architecture, which gives us a distinct advantage compared to existing methods. For a fair comparison, all RGB inputs were resized to half their original resolution, matching the scale of our RAW inputs (which are converted to four-channel inputs at half resolution based on the bayer pattern).
>
> The table below reports the average inference time, parameters for processing single-frame 7028×4688 RAW images (Ours) or 3514×2344 RGB images (Other Methods) on the same GPU:
>
> |    Method    | Inference Time(s) | Parameters(M) |
> | :----------: | :---------------: | :-----------: |
> | HDRTVNet [3] |       1.09        |     1.59      |
> | RetinexMamba [4] |       1.58        |     4.41      |
> | FluxFill [5] |      142.13       |       —       |
> | UltraFusion  |      115.26       |    1774.08    |
> |     Ours     |       1.07        |     16.24     |
>
> As shown in the table, owing to the relatively simple architecture of our network, our method achieves a faster inference speed compared to HDRTVNet and RetinexMamba, and significantly outperforms UltraFusion and FluxFill in this regard. While the parameters in our model is substantially lower than that of UltraFusion, it is larger than those of RetinexMamba and HDRTVNet. Nonetheless, considering the superior performance achieved by our method, the increase in parameters is well justified.
>
>
> # 3 Discussion on the Fundamental Value of Single-Frame Methods
>
>  Beyond the inherent limitations of multi-frame methods—such as their sensitivity to motion and difficulty in aligning dynamic content—single-frame approaches possess unique and irreplaceable advantages in a range of real-world applications. For instance, scenarios involving flash photography, high-speed imaging, or event-driven capture often permit only a single exposure, making multi-frame acquisition infeasible or even impossible.
>
> It is therefore inappropriate to dismiss the significance of single-frame methods simply because multi-frame alternatives may perform well under controlled or static conditions. Single-frame approaches address a fundamentally different set of practical constraints, and the ability to reconstruct high-quality results from a single short-exposure input is essential for enabling reliable performance in time-critical or motion-rich environments.
>
> In this context, our work not only contributes to the advancement of single-frame HDR reconstruction from RAW images, but also reinforces the broader relevance and necessity of research in this direction.
>
>
>
> [1] UltraFusion: Ultra High Dynamic Imaging using Exposure Fusion
>
> [2] Physics-based Noise Modeling for Extreme Low-light Photography
>
> [3] A New Journey from SDRTV to HDRTV
>
> [4] Retinexmamba: Retinex-based Mamba for Low-light Image Enhancement
>
> [5] Flux. https://github.com/black-forest-labs/flux

---

> ### Author Response · Authors · 2025-08-06
>
> Dear Reviewer N566,
>
> Thank you very much for your assessment and valuable suggestions.
>
> If you have any further comments or require additional clarification, we would greatly appreciate your continued feedback.
>
> Thank you again for your time and review.
>
> Best regards,
> The authors

---

> ### Author Response · Authors · 2025-08-08
>
> Dear Reviewer N566,
>
> I hope this message finds you well. As there is only one day remaining before the discussion period concludes, we would like to express our sincere gratitude for your valuable suggestions. We have provided targeted responses to address your comments and would greatly appreciate it if you could review them to see whether they resolve your concerns.
>
> If there are any remaining issues, please do let us know. Your insights are of great importance to us, and we would be glad to engage in further discussion or provide additional clarification.
>
> Thank you for your time and thoughtful feedback.
>
> Best regards,
>
> The Authors

---

### Note · Authors · 2025-08-11

First, we would like to thank the AC and all reviewers for their time.

We are delighted to have had the opportunity to engage in discussions with the reviewers, during which many of them gained a clearer understanding of our work's novelty and the clarifications we provided. Reviewer 2iPX expressed strong recognition of our work, while both Reviewer UcfZ and Reviewer e9Nm agreed to raise their scores to positive values. Although Reviewer N566 did not respond during the discussion phase—possibly due to unavoidable circumstances—the concerns raised were in fact a subset of those from Reviewer e9Nm, and our corresponding clarifications have already been acknowledged and accepted by e9Nm.

We believe our work is well-suited for acceptance at NeurIPS because we are the first to leverage a short-exposure RAW image for HDR reconstruction. Furthermore, we are the first to jointly address the challenges of denoising and HDR tone mapping from short-exposure RAW input—a long-standing problem highlighted by SID [1] but left unresolved for years. Specifically, our contributions are as follows:

1) We propose a novel method for reconstructing UHDR scenes from a single-frame RAW image, introducing a two-stage pipeline that decouples exposure correction from denoising, fully exploiting the properties and advantages of RAW data.

2) We introduce a brightness-aware noise model and a ratio-map encoding scheme that work synergistically to guide the network in recovering fine details across varying exposure levels.

3) We design a new data pipeline for multi-exposure fusion and contribute a corresponding dataset for benchmarking UHDR reconstruction performance.

Finally, we will incorporate all reviewer suggestions in the final version to further enhance the quality of our paper.

[1] Learning to See in the Dark

---

### Decision · Program_Chairs · 2025-09-17

**Decision:**

Accept (poster)

**Comment:**

This paper received four mixed, but largely positive-leaning reviews.

There was general appreciation for the motivation and significance of the problem considered here, the technical soundness and practicality of the approach, and the quality of the results.


There were some concerns raised in the initial reviews regarding experimental evaluation (strength of baseline comparisons) and presentation quality. There was significant author-reviewer discussions following the initial review period, after which three of the reviewers converged to an accept rating (1 Strong Accept, 2 Borderline Accepts). The fourth reviewer (with an initial borderline reject rating) did not participate in the discussions, nor updated their final rating or respond to the author rebuttal.

Therefore, on balance, an accept decision was reached.